# A 3D culture model of innervated human skeletal muscle enables studies of the adult neuromuscular junction

Mohsen Afshar Bakooshli[1,2], Ethan S Lippmann[3,4], Ben Mulcahy[5], Nisha Iyer[3,4], Christine T Nguyen[6], Kayee Tung[7], Bryan A Stewart[6,8], Hubrecht van den Dorpel[2,9], Tobias Fuehrmann[1,10], Molly Shoichet[1,2,10], Anne Bigot[11], Elena Pegoraro[12], Henry Ahn[7,13], Howard Ginsberg[2,7,13], Mei Zhen[5,14,15], Randolph Scott Ashton[3,4], Penney M Gilbert[1,2,6,16]*

[1]Donnelly Centre, University of Toronto, Toronto, Canada; [2]Institute of Biomaterials and Biomedical Engineering, University of Toronto, Toronto, Canada; [3]Department of Biomedical Engineering, University of Wisconsin-Madison, Madison, United States; [4]Wisconsin Institute for Discovery, University of Wisconsin-Madison, Madison, United States; [5]Lunenfeld-Tanenbaum Research Institute, Mount Sinai Hospital, Toronto, Canada; [6]Department of Cell and Systems Biology, University of Toronto, Toronto, Canada; [7]Department of Surgery, University of Toronto, Toronto, Canada; [8]Department of Biology, University of Toronto Mississauga, Mississauga, Canada; [9]Department of Pharmaceutics, Utrecht University, Utrecht, Netherlands; [10]Department of Chemical Engineering and Applied Chemistry, University of Toronto, Toronto, Canada; [11]INSERM, Association Institut de Myologie, Centre de Recherche en Myologie, Sorbonne Universite, Paris, France; [12]Department of Neuroscience, University of Padova, Padova, Italy; [13]Li Ka Shing Knowledge Institute, St. Michael's Hospital, Toronto, Canada; [14]Department of Physiology, University of Toronto, Toronto, Canada; [15]Department of Molecular Genetics, University of Toronto, Toronto, Canada; [16]Department of Biochemistry, University of Toronto, Toronto, Canada

*For correspondence:
Penney.Gilbert@utoronto.ca

**Competing interests:** The authors declare that no competing interests exist.

**Abstract** Two-dimensional (2D) human skeletal muscle fiber cultures are ill-equipped to support the contractile properties of maturing muscle fibers. This limits their application to the study of adult human neuromuscular junction (NMJ) development, a process requiring maturation of muscle fibers in the presence of motor neuron endplates. Here we describe a three-dimensional (3D) co-culture method whereby human muscle progenitors mixed with human pluripotent stem cell-derived motor neurons self-organize to form functional NMJ connections. Functional connectivity between motor neuron endplates and muscle fibers is confirmed with calcium imaging and electrophysiological recordings. Notably, we only observed epsilon acetylcholine receptor subunit protein upregulation and activity in 3D co-cultures. Further, 3D co-culture treatments with myasthenia gravis patient sera shows the ease of studying human disease with the system. Hence, this work offers a simple method to model and evaluate adult human NMJ de novo development or disease in culture.
DOI: https://doi.org/10.7554/eLife.44530.001

## Introduction

The skeletal muscle neuromuscular junction (NMJ) is a highly organized synapse formed between a motor neuron (MN) axon and a muscle fiber. It is designed to transmit efferent signals from projecting MNs to muscle fibers in order to actuate fiber contraction. Nicotinic acetylcholine receptors (AChRs) clustered at the NMJ's postsynaptic muscle fiber membrane mediate this signal by binding acetylcholine (ACh) neurotransmitters released from vesicles at the presynaptic MN axon terminal. AChRs are ligand-gated ion channels composed of five protein subunits. During development the gamma subunit in embryonic AChRs is replaced by an epsilon subunit in the adult synapse (*Mishina et al., 1986*; *Missias et al., 1996*). Previous animal studies showed that this AChR subunit transition occurs in the presence of motor axon endplates and confirmed that transcription of the epsilon gene (CHRNE) is stimulated by AChR Inducing Activity (ARIA) via ErbB receptors, a nerve derived ligand of the neuregulin-1 (NRG1) family (*Martinou et al., 1991*). Consistently, CHRNE transcripts are detected in rodent 2D and 3D skeletal muscle fiber cultures when co-cultured with nerve cells (*Bach et al., 2003*; *Ostrovidov et al., 2017*; *Smith et al., 2016*; *Vilmont et al., 2016*). However, despite significant progress toward directing human pluripotent stem cells (PSCs) to the motor neuron lineage (*Ashton et al., 2015*; *Hu and Zhang, 2010*; *Lippmann et al., 2014*; *Maury et al., 2015*; *Shimojo et al., 2015*; *Zhang et al., 2001*) and establishing electrically and chemically responsive human muscle fibers in vitro (*Madden et al., 2015*), the first reports of human NMJ models – 2D (*Guo et al., 2011*; *Santhanam et al., 2018*; *Steinbeck et al., 2016*) or 3D (*Maffioletti et al., 2018*; *Osaki et al., 2018*) human muscle fiber and motor neuron co-cultures – do not demonstrate synapse maturation via the gamma to epsilon AChR subunit switch. Further, there are no reports of epsilon AChR protein expression or function in culture in the absence of enforced gene expression.

Congenital myasthenic syndrome is one of the most prevalent genetic diseases of the NMJ and commonly arises from mutations in one of the AChR encoding genes (*Engel et al., 2010*). The vast majority of mutations causing the disease arise in the CHRNE gene, the adult specific subunit of the AChR (*Abicht et al., 2012*; *Engel et al., 1993*). Given the lack of effective therapies for a wide range of neuromuscular diseases impacting the adult NMJ (*Ohno et al., 1999*), and that the majority of AChR mutations are mutations of the CHRNE gene (*Ohno et al., 1995*), a robust method to model the adult human NMJ in a dish is needed to synergize with recent advances in differentiating patient-derived PSCs to the MN lineage (*Chen et al., 2011*; *Hu et al., 2010*; *Lorenz et al., 2017*; *Sances et al., 2016*).

Here we report a method integrating architectural cues with co-culture techniques to create an environment conducive to the de novo formation of the adult human NMJ in as early as two weeks. In side-by-side studies of muscle fibers cultured in 2D, we show that the 3D culture system enables long-term maintenance of maturing muscle fibers in culture. It supports the formation and morphological maturation of AChR clusters primed for synaptogenesis and the de novo transition from the embryonic to the adult NMJ composition upon contact with MN endplates. We confirm formation of functional NMJ connections by imaging muscle fiber calcium transients and capturing electrophysiological recordings in response to glutamate-induced MN firing and demonstrate that treatment with inhibitors targeting pre- and post-synapse function block this firing. We show that the 3D co-culture platform, and not a 2D co-culture system, supports the transition from the embryonic to the adult AChR, thereby enabling the functional assessment of the adult neuromuscular junction in vitro. We present data aligning with prior studies showing that epsilon functional activity is regulated post-transcriptionally (*Bruneau et al., 2005*; *Caroni et al., 1993*; *Jayawickreme and Claudio, 1994*; *Khan et al., 2014*; *Missias et al., 1996*; *Ross et al., 1991*; *Wild et al., 2016*; *Witzemann et al., 2013*; *Xu and Salpeter, 1997*; *Yampolsky et al., 2008*), and in particular, supports work indicating a role for innervation (spontaneous miniature endplate potentials) and/or muscle fiber maturation in encouraging subunit substitution (*Caroni et al., 1993*; *Missias et al., 1996*; *Witzemann et al., 2013*; *Xu and Salpeter, 1997*; *Yampolsky et al., 2008*). Finally, we demonstrate the versatility and ease of the system for modeling human disease by treating neuromuscular co-cultures with IgG purified from myasthenia gravis (MG) patient sera together with human complement, which results in readily visible clinical-like phenotypes in as early as two weeks of culture time. Thus, the described 3D co-culture model provides a method to investigate adult human NMJ development, and therefore adult forms of neuromuscular diseases, in vitro for the first time.

## Results

### Myogenic differentiation in 3D enhances fiber maturation and AChR clustering

We performed a side-by-side comparison of human skeletal muscle fiber populations derived in standard 2D culture versus 3D culture and uncovered differences in fiber maturation and AChR clustering (*Figure 1—figure supplement 1A*). We established primary myogenic progenitor and fibroblast-like cell lines from human biopsy tissues (*Blau and Webster, 1981*) (*Figure 1—figure supplement 1B*), and seeded them at defined ratios either within a fibrin/Geltrex hydrogel (3D) or into 12-well tissue culture plastic dishes coated with Geltrex (2D) or a fibrinogen/Geltrex blend (*Figure 1—figure supplement 1A*). Muscle cell laden hydrogels were formed within a polydimethylsiloxane channel and anchored at each end of the channel to the nylon hooks of Velcro fabric (*Bell et al., 1979*; *Madden et al., 2015*; *Vandenburgh et al., 1988*), which act as artificial tendons and establish uniaxial tension during 3D tissue remodeling and differentiation (*Figure 1—figure supplement 1C*).

Immunofluorescence analysis of the muscle contractile protein sarcomeric α-actinin (SAA) revealed the uniform alignment of striated muscle fibers along the tension axis in the 3D tissues (*Figure 1A* and *Figure 1—figure supplement 1E*), while 2D muscle fiber cultures were regionally aligned (*Figure 1A*), but globally disorganized (*Figure 1—figure supplement 1D*). In contrast to the muscle fibers established in 2D cultures, those derived in 3D culture progressively increased in diameter over three weeks in culture (*Figure 1B*) while maintaining fiber alignment and assembled contractile apparatus (*Figure 1A*). Furthermore, over time in 3D culture, muscle tissues upregulated expression of the fast and slow adult isoforms of myosin heavy chain (MHC), which was accompanied by a downregulation of embryonic MHC expression, suggesting a gradual sarcomere structural maturation (*Figure 1C* and *Figure 1—figure supplement 2A–E*). The absence of these trends in 2D muscle fiber culture may be explained by the inability of tissue culture plastic to support muscle fiber contraction resulting in the increased incidence of damaged fibers observed in 2D cultures (*Figure 1—figure supplement 1D*) and an enrichment of small, immature fibers (*Figure 1A–B* and *Figure 1—figure supplement 2G–H*).

In support of our molecular characterization, 3D human muscle tissues were capable of generating active force in as early as 10 days of differentiation as evidenced by spontaneous twitches (*Video 1*), which were not observed in 2D cultures. Consistent with prior reports (*Madden et al., 2015*), two-week old 3D muscle tissue twitch response could be paced by low frequency electrical stimuli (1 Hz; *Video 1*), which converted into tetanus contractions in response to increased frequency (20 Hz; *Video 1*). Similarly, ACh stimulation (10 μM) produced an immediate tetanus response (*Video 1*) in 3D tissues suggesting an abundance of active AChRs, while the response of 2D muscle fiber cultures at this time-point was significantly less and inevitably resulted in muscle fiber damage and/or release from the culture substrate (*Video 2*).

To evaluate the calcium handling capacity of 3D muscle fiber cultures, we transduced human muscle progenitor cells with lentiviral particles encoding GCaMP6 (*Chen et al., 2013*), a sensitive calcium indicator protein, driven by the MHCK7 (*Madden et al., 2015*) promoter, a muscle specific gene. Muscle fibers in 3D tissues generated strong collective calcium transient in response to electrical stimulation and immediately following exposure to ACh (*Figure 1—figure supplement 3A–C* and *Video 3*).

To evaluate the electrophysiological characteristics of single muscle fibers in 3D cultures, muscle progenitor cells were stably transduced with a light-gated ion channel, channelrhodopsin-2 (ChR2), driven by an EF1α promoter (*Zhang et al., 2007*). 3D muscle tissues generated using optogenetically-responsive muscle progenitor cells contracted in response to light stimulation on the second week of the culture (*Video 4*). Single muscle fiber membrane potentials were recorded in these tissues using sharp microelectrode recording (*Figure 1—figure supplement 3D*). As expected, recordings of 3D muscle prior to light stimulation revealed little electrical activity (*Figure 1—figure supplement 3E*), while light activation generated a clear depolarization of the membrane potential (*Figure 1—figure supplement 3F*). We also took a more traditional approach using single, sharp electrode electrophysiology to measure membrane potential and test excitability. Passing depolarizing current led to regenerative potentials that become faster with increasing depolarization (*Figure 1—figure supplement 3G*).

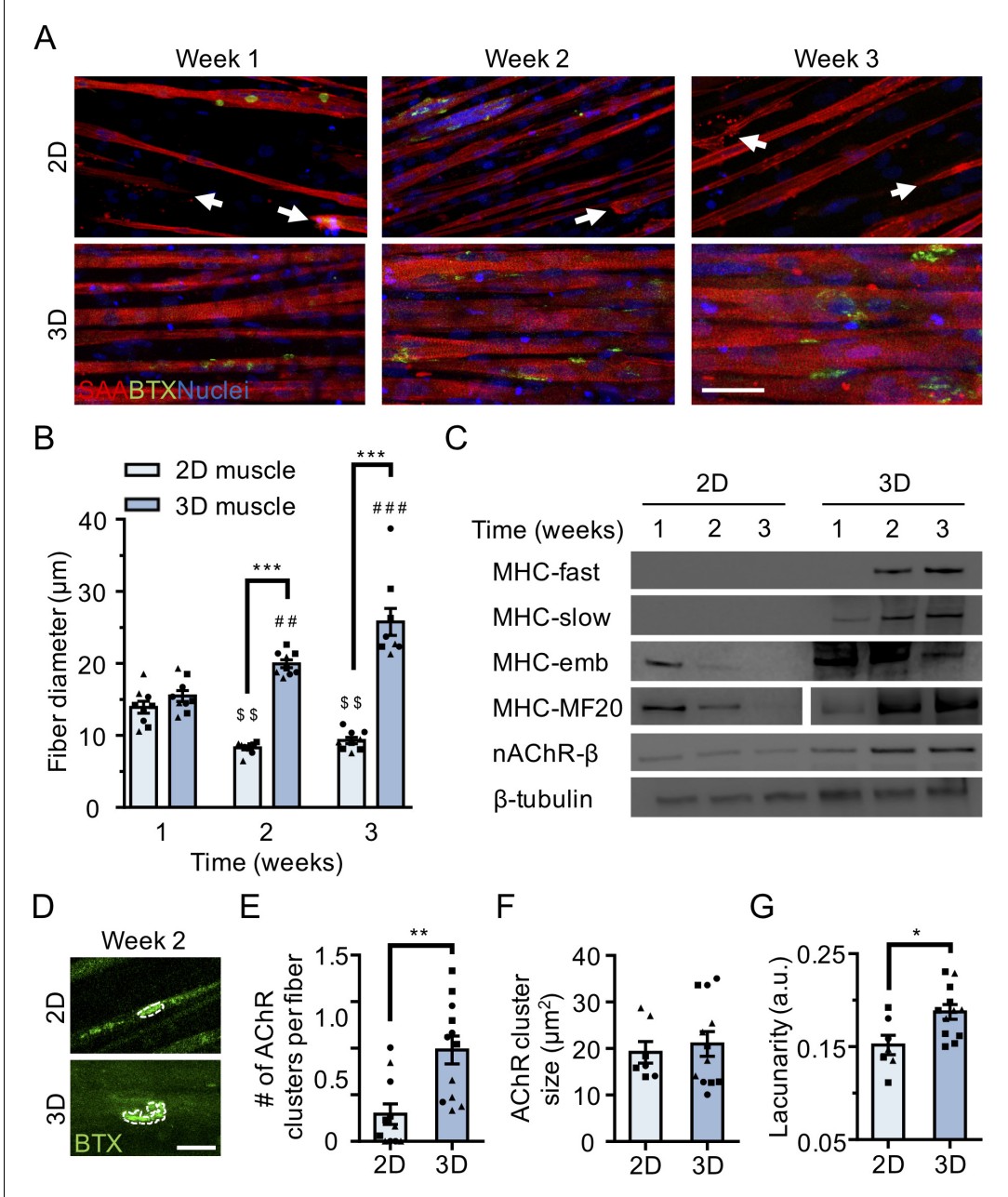

**Figure 1.** 3D culture enhances skeletal muscle fiber maturation over 2D culture. (**A**) Representative confocal images of muscle fibers established in 2D (top row) and 3D conditions and immunostained for sarcomeric α-actinin (SAA; red), α-bungarotoxin (BTX; green), and Hoechst 33342 (blue) after 1, 2, and 3 weeks of culture. Scale bar, 50 μm. White arrowheads indicate broken fibers. (**B**) Bar graph of muscle fiber diameter quantified in 2D (light blue) and 3D (blue) cultures over time. n = 9 independent samples from three muscle patient donors. A minimum of 50 myotubes per time point per patient sample were analyzed. ##p<0.01 and ###p<0.001 compared with 3D cultures at week 1. $$p<0.01 compared with 2D culture at week 1. (**C**) Representative western blot images of myosin heavy chain (MHC) isoforms (fast, slow, embryonic (emb), and pan (MF-20)) nicotinic AChR-β (nAChR-β), and β-tubulin in 2D compared with 3D cultures over time. (**D**) Representative confocal images of muscle fibers cultured in 2D or 3D for two weeks and then labeled with α-bungarotoxin (green). AChR clusters are outlined with white dashed lines. Scale bar, 25 μm. (**E–G**) Bar graphs indicating average (**E**) number of AChR clusters per fiber, (**F**) AChR cluster size, and (**G**) AChR cluster lacunarity in 2D (light blue) and 3D (blue) muscle fiber cultures at week 2. n = minimum of 9 independent samples from three muscle patient donors. A minimum of 30 microscopic images per culture condition were analyzed. In (**B**), (**C**), and (**E–G**) each symbol represents data from one muscle patient donor. Values in (**B**), (**E**), (**F**), and (**G**) are mean ±SEM. *p<0.05, **p<0.01, ***p<0.001.

DOI: https://doi.org/10.7554/eLife.44530.002

The following figure supplements are available for figure 1:

*Figure 1 continued on next page*

*Figure 1 continued*

**Figure supplement 1.** Two- and three-dimensional methods to culture human muscle fibers.
DOI: https://doi.org/10.7554/eLife.44530.003
**Figure supplement 2.** Comparison of muscle fiber maturation in 2D and 3D cultures.
DOI: https://doi.org/10.7554/eLife.44530.004
**Figure supplement 3.** Functional characterization of 3D skeletal muscle tissues.
DOI: https://doi.org/10.7554/eLife.44530.005

Finally, we compared AChR clustering, an integral step in NMJ development, in 2 week differentiated 2D and 3D muscle fiber cultures (). We observed significantly higher expression of the nAChR-β protein in 3D compared to 2D cultures at 2 weeks of fiber differentiation (*Figure 1C* and *Figure 1—figure supplement 2A* and *2F*). Further, our analyses revealed a greater number of AChR clusters per muscle fiber established in 3D compared to 2D culture (*Figure 1E*). Indeed, we noted that at 2 weeks of culture, the majority of muscle fibers in 2D cultures lacked AChR clusters (*Figure 1A and E*). Interestingly, although average AChR cluster area was not significantly different (*Figure 1F*), we observed a high frequency of branched and perforated AChR clusters in our 3D muscle cultures, whereas oval shaped AChR clusters dominated on muscle fibers cultured in 2D conditions (*Figure 1D*). To quantify this observation, we assessed the lacunarity of AChR clusters formed on muscle fibers cultured in 2D and 3D conditions. Lacunarity is a measure of shape morphological heterogeneity and 'gappiness'. Patterns with high lacunarity contain gaps or 'lacunas', whilst lower lacunarity implies pattern homogeneity or rotational invariance (*Karperien et al., 2013*; *Smith et al., 1996*). Lacunarity calculated from box counting validated our qualitative observations by indicating a significantly higher average lacunarity of AChR clusters formed in 3D cultures compared to 2D cultures (*Figure 1G*).

Overall our comparison of muscle fibers established in 2D and 3D formats suggests that the 3D culture method better supports rapid contractile apparatus maturation and function, as well as AChR clustering and morphological maturation.

## 3d human neuromuscular co-cultures recapitulate early NMJ synaptogenesis

Since muscle fiber maturation is a prerequisite for NMJ development (*Fox, 2009*), we evaluated the hypothesis that the 3D skeletal muscle tissue platform would be well suited for human PSC-derived MN incorporation to model human NMJ synaptogenesis. We utilized MN clusters (Day 20) differentiated from WA09 human embryonic stem cell – derived OLIG2$^+$ progenitor cells (*Figure 2—figure supplement 1A–B*) (*Lippmann et al., 2015*). Resulting MN clusters were enriched (>85%) for cells expressing the HB9 and ISL1 transcription factors as well as the mature neurofilament marker SMI32 (*Figure 2—figure supplement 1C–D*). MN clusters were collected prior to muscle tissue preparation, mixed with the muscle progenitor cells in the hydrogel mix, and seeded together into the PDMS channels. The 3D skeletal muscle tissue media was optimized to support co-culture health by supplementation with brain derived and glial cell line derived neurotrophic factors (BDNF, GDNF) to support MN viability.

Co-cultures examined after 10 days in differentiation media showed close contact between the MN clusters and the muscle tissue by phase-contrast microscopy (*Figure 2A*). Immunostaining co-cultures on the second week of culture for the motor neuron marker SMI-32, muscle fiber marker sarcomeric α-actinin, and α-bungarotoxin

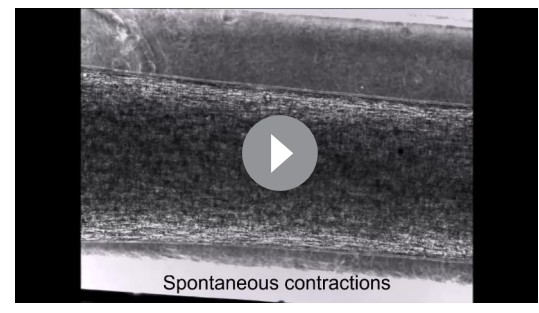
Spontaneous contractions

**Video 1.** Three-dimensional human skeletal muscle tissue contraction in response to chemical and electrical stimulation. A series of four representative bright-field real-time videos of three-dimensional human muscle tissues after 10–12 days of culture exhibiting spontaneous contractions, or contracting in response to electrical (1 Hz, 20 Hz) or acetylcholine (10 μM) stimulation.
DOI: https://doi.org/10.7554/eLife.44530.006

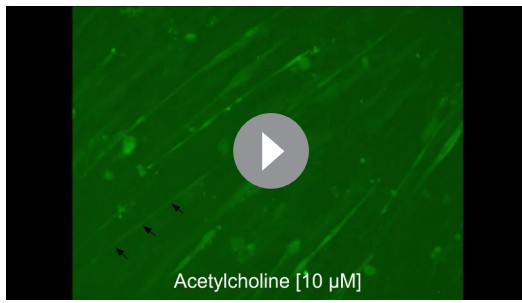

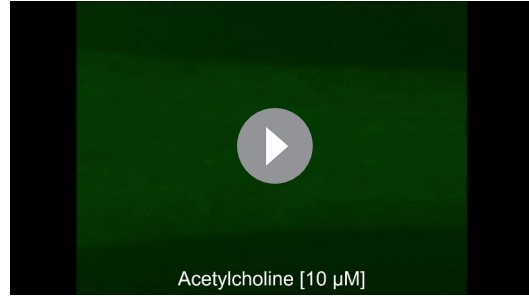

**Video 2.** Two-dimensional human skeletal muscle fiber contraction in response to acetylcholine stimulation. Epifluorescence real-time video of a two-dimensional GCaMP6 transduced human muscle fiber culture contracting in response to 10 µM acetylcholine stimulation after 2 weeks of culture. Black arrow heads indicate a muscle fiber that breaks post acetylcholine stimulation.
DOI: https://doi.org/10.7554/eLife.44530.007

**Video 3.** Three-dimensional human skeletal muscle tissue calcium handling in response to chemical stimulation. A series of two representative epifluorescence time-lapse videos of three-dimensional human skeletal muscle tissues after 10–12 days of culture stimulated with acetylcholine (10 µM) and then with L-glutamate (50 µM). Muscle fiber calcium transients are visualized in green by following a GCaMP6 calcium reporter that was transduced into the human muscle cells.
DOI: https://doi.org/10.7554/eLife.44530.008

(to visualize AChRs) revealed that the co-cultures self-organized such that muscle progenitor cells fused to form multinucleated, aligned and striated muscle fibers and the MN clusters were positioned at the periphery of muscle bundles (*Figure 2B*). Importantly, the MNs were capable of regrowing neurites that were found in contact with α-bungarotoxin positive AChR clusters on muscle fibers (*Figure 2B–C*). In vivo studies by others found that postsynaptic AChR aggregation on muscle fibers is supported by agrin secretion from MN axon terminals (*Gautam et al., 1996*). We confirmed agrin expression in our PSC-derived MN cultures (*Figure 2—figure supplement 1E*). Furthermore, western blot analysis of neuromuscular co-cultures confirmed expression of MuSK (*Figure 2—figure supplement 2A*) and rapsyn (*Figure 2—figure supplement 2B*) proteins, two decisive synaptic proteins for mediating agrin-induced synaptogenesis (*Glass and Yancopoulos, 1997*). We did not observe any examples of bungarotoxin-labeled AChR clusters co-localizing with either rapsyn or MuSK proteins in our 2D muscle-alone or neuromuscular co-cultures (*Figure 2—figure supplement 2C*). In our study we observed a single incidence of MuSK protein co-localization with an AChR cluster in our 3D muscle-alone cultures (*Figure 2—figure supplement 2D*) and no examples of rapsyn co-localization with AChR clusters. By comparison, the prevalence of rapsyn (*Figure 2D*) and MuSK (*Figure 2—figure supplement 2E*) co-localization with bungaroxin-labeled AChR clusters was substantially higher in 3D neuromuscular co-cultures, but was not observed on all fibers.

Consistently, we observed more and larger α-bungarotoxin positive AChR clusters in 3D neuromuscular co-cultures as compared to 3D muscle-alone cultures (*Figure 2E–G*), particularly at sites where MN neurites contacted muscle fibers (*Figure 2B*, yellow boxes). In addition, we observed a higher frequency of perforated and branched AChR clusters in our 3D co-cultures as evidenced by the higher lacunarity of AChR clusters formed in co-cultures (*Figure 2H*) supporting a role for motor axon derived factors in post synaptic differentiation of the NMJ. As expected, by supplementing 3D human muscle

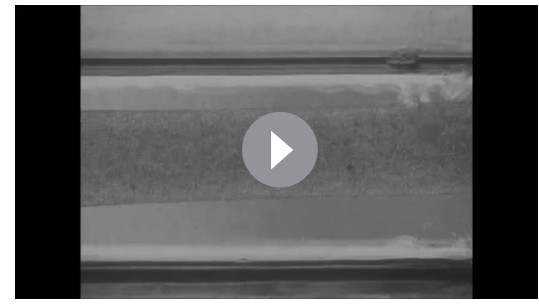

**Video 4.** Optogenetically transduced three-dimensional human skeletal muscle tissue response to blue light (470 nm). A real-time bright field video of a 3D human skeletal muscle tissue transduced with ChR2 (H134R) and stimulated by blue light. Red circles indicate the time and period of light pulses.
DOI: https://doi.org/10.7554/eLife.44530.009

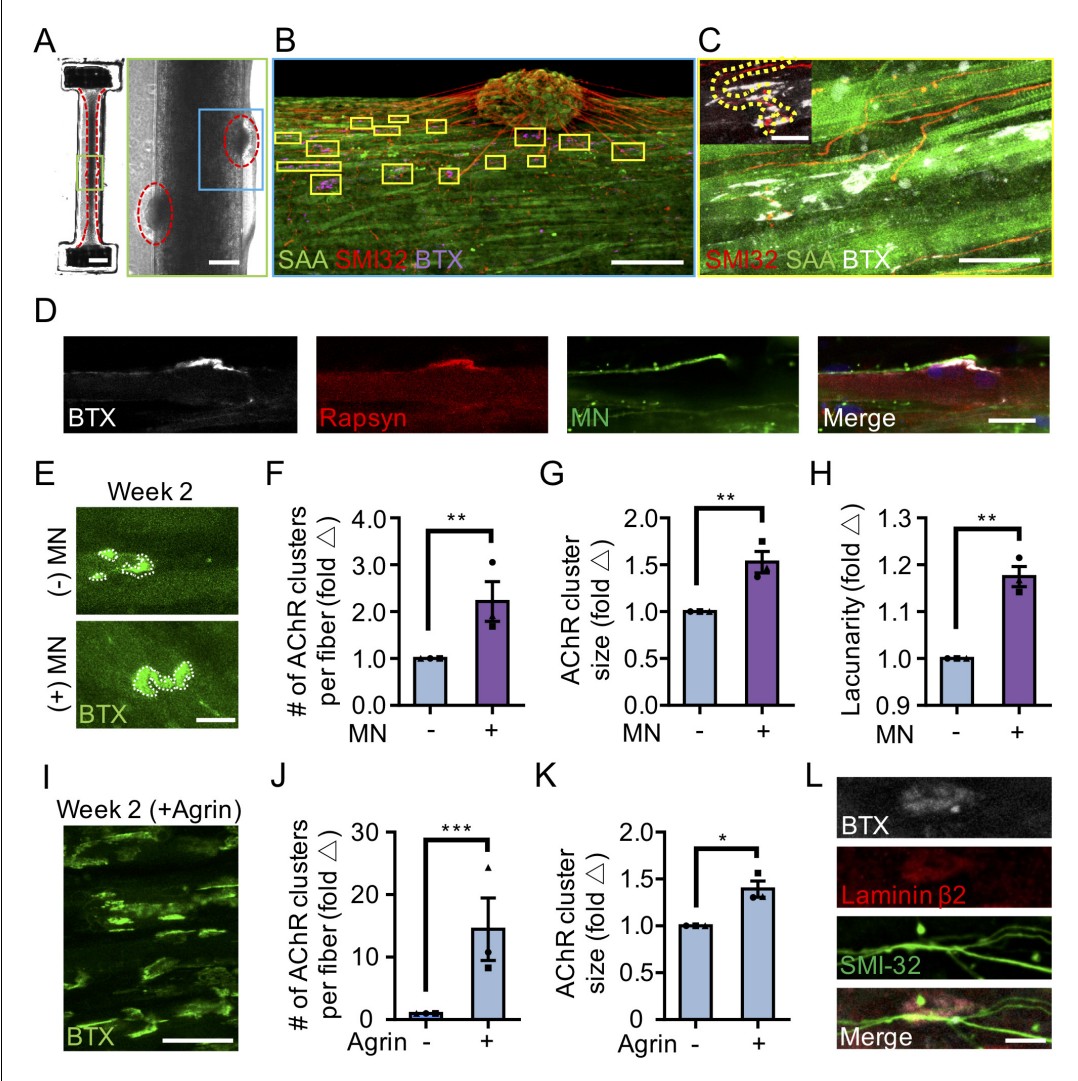

**Figure 2.** 3D neuromuscular co-culture augments AChR clustering and maturation. (**A**) Stitched phase contrast image of a representative 3D skeletal muscle-motor neuron (MN) co-culture at two weeks of culture. Neuromuscular tissue outlined with red dashed line in left panel. Region outlined in green box is magnified in the image to the immediate right. Red dashed lines in right panel outline motor neuron clusters. Scale bars, 2 mm (left panel) and 200 μm (right panel). (**B**) Representative confocal image of a two-week old neuromuscular co-culture immunostained for sarcomeric α-actinin (SAA; green), α-bungarotoxin (BTX; magenta), and neurofilament heavy SMI-32 (red). AChR clusters co-localized with neurites are outlined with yellow boxes. Scale bar, 200 μm. (**C**) Representative confocal image indicating co-localization of a SMI-32 (red) labeled neurite terminal and a BTX (white) labeled AChR cluster on a striated muscle fiber as seen by SAA (green) staining. Scale bar, 50 μm. (**D**) Representative confocal image of a neuromuscular co-culture immunostained on Day 10 of differentiation for Rapsyn (red), bungarotoxin (BTX, white), and counter stained with Hoechst 33342 to visualize the nuclei (blue). Motor neurons (green) were derived from GFP expressing human iPSCs. Scale bar 25 μm. (**E,I**) Representative confocal images of AChR clusters formed on muscle fibers cultured in 3D (**E**) with (+) or without (-) motor neurons (MN) or (**I**) supplemented with agrin and labeled with α-bungarotoxin after two weeks of culture. Scale bars, 25 μm (**E**) and 50 μm (**I**). AChR clusters are outlined with white dashed lines in (**E**). (**F–H, J–K**) Bar graphs indicating average (**F,J**) number of AChR clusters per fiber, (**G,K**) AChR cluster size, and (**H**) AChR cluster lacunarity in 3D cultures (**F–H**) with (+; purple) or without (-; blue) MN or (**J–K**) with or without agrin supplementation at week 2. In (**F–H**), values are normalized to 3D muscle cultures without MNs. In (**J–K**) values are normalized to untreated control. (**L**) Representative confocal image of a neuromuscular co-culture immunostained for laminin-β 2 (red), bungarotoxin (BTX, white), and SMI-32 (green). Scale bars, 10 μm. For (**F–H**) and (**J–K**), n = minimum of 9 independent samples from three muscle patient donors. For agrin treated samples in (**J–K**), 6 samples from three muscle donors were analyzed. A minimum of 30 (**F–H**) or 6 (**J–K**) microscopic images per culture condition were analyzed. In (**F–H**) and (**J–K**) each symbol represents data from one muscle patient donor. Values in (**F–H**) and (**J–K**) are mean ±SEM. *p<0.05, **p<0.01, and ***p<0.001.

DOI: https://doi.org/10.7554/eLife.44530.010

The following figure supplements are available for figure 2:

**Figure supplement 1.** Generation and basic characterization of ESC-derived motor neurons.

*Figure 2 continued on next page*

*Figure 2 continued*

DOI: https://doi.org/10.7554/eLife.44530.011

**Figure supplement 2.** 2D and 3D neuromuscular co-culture characterization.

DOI: https://doi.org/10.7554/eLife.44530.012

tissue media with neural agrin (50 ng/mL) we phenocopied these co-culture results (*Figure 2I–K*). An evaluation of 2D neuromuscular co-cultures at the same time-point revealed a local alignment of the neurites and muscle fibers (*Figure 2—figure supplement 2C and F*, right panel), and a qualitative improvement in MN health and muscle fiber number and integrity (data not shown). However, only rare muscle fibers possessed clustered AChRs and we could not detect co-localization of the AChRs with SMI-32 stained neurites at this time point (*Figure 2—figure supplement 2F*).

In further support of 3D muscle fiber synaptogenic maturation, the LAMB2 gene encoding for the laminin beta two chain was expressed by 3D human muscle tissues and neuromuscular co-cultures (*Figure 2—figure supplement 2G*), and the protein was found enriched at AChR clusters (*Figure 2L*). This is consistent with prior reports demonstrating laminin beta two concentrated at the neuromuscular junction synaptic cleft (*Hunter et al., 1989*) and the involvement of this tissue restricted basement membrane protein in NMJ maturation and maintenance (*Noakes et al., 1995*).

Our characterizations demonstrate that a 3D neuromuscular co-culture system recapitulates many aspects of early synaptogenesis that were first identified with in vivo studies.

## 3d human neuromuscular co-cultures are functionally innervated

We next sought to evaluate NMJ functionality in our neuromuscular co-cultures. With a combination of calcium handling analyses and electrophysiological recordings we report that 3D human neuro-muscular co-cultures are functionally innervated in as early as two weeks. Using the fluorescent styryl dye FM 1–43 (*Gaffield and Betz, 2006*) and confocal microscopy we performed exocytosis assays on differentiated MNs (Day 20) and confirmed that human PSC-derived MNs exocytose in response to potassium chloride (KCl, 60 mM) and the excitatory neurotransmitter, L-glutamate (50 μM) stimuli (*Figure 3—figure supplement 1A–D* and *Video 5*). The latter is particularly important, since the amino acid glutamate is a neurotransmitter that specifically stimulates MN cells but not muscle fibers (50 μM; *Video 3*).

Next, we stimulated neuromuscular co-cultures that were generated using GCaMP6 transduced muscle progenitor cells with a 50 μM glutamate solution and observed calcium transients (*Figure 3A–B* and *Video 6*) and synchronous tissue contractions (*Figure 3C* and *Video 7*) in the mus-cle fibers in close proximity to the MN clusters in as early as 14 days of co-culture, indicating the formation of functional connectivity between MN endplates and muscle fibers. Stimulating the same tissue with ACh, following the glutamate stimulation, provided a rapid way to stimulate and visualize all muscle fibers in the tissue. Our analysis of this serial stimulation data revealed that many, but not all the fibers were functionally innervated (*Figure 3A–B* and *Video 6*). As expected, direct stimulation of AChRs using ACh led to higher co-culture contractile force genera-tion as quantified by tissue movement (*Figure 3C*). To further validate that the presyn-aptic activation of motor neurons (i.e. glutamate stimulation) caused the observed changes in muscle fiber calcium transients and muscle fiber contractions, we studied the effect of BOTOX (BOT, presynaptic blocker) and d-tubocurarine (DTC, post synaptic blocker) treatments in our system. Our studies revealed a significant

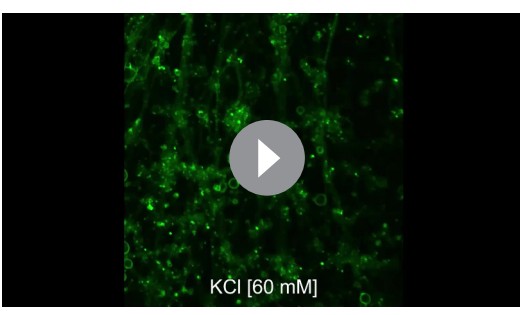

**Video 5.** Pluripotent stem cell derived motor neurons exocytose in response to physiological stimuli. Representative time-lapse microscope videos of FM 1–43 loaded PSC-derived motor neurons response to various stimuli (KCl (60 mM), HBSS and L-Glutamate (50 μM)). Videos illustrate loss of fluorescent intensity from the neurites post glutamate and potassium chloride stimuli indicative of exocytosis. White lines outline the neurites in HBSS and L-Glutamate stimulation videos.

DOI: https://doi.org/10.7554/eLife.44530.013

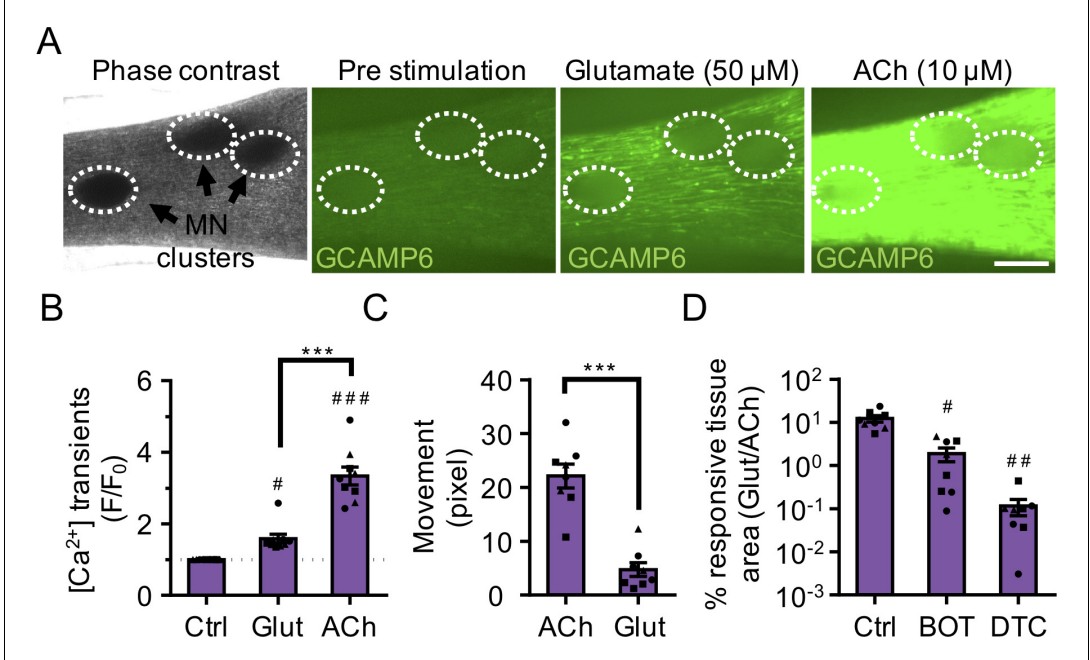

**Figure 3.** 3D neuromuscular co-cultures are functionally innervated. (A) Phase contrast (far left panel) and GCaMP6 epifluorescence images (right panels) of a 3D neuromuscular co-culture after treatment with phosphate buffered saline (middle left panel), glutamate (middle right panel), or ACh (far right panel). Motor neuron clusters are outlined with white dashed lines. Scale bar, 250 µm. (B) Bar graph indicating quantification of fluorescence signal from neuromuscular co-cultures following glutamate (Glut) and Acetylcholine (ACh) stimulations relative to treatment with phosphate buffered saline (Ctrl). n = 9 neuromuscular co-culture samples from three separate muscle patient donors. #p<0.05 and ###p<0.001 compared with saline stimulation (Ctrl). (C) Quantification of neuromuscular co-culture tissue contraction in response to ACh (10 µM) and glutamate (50 µM). (D) Bar graph quantification of the percent tissue area occupied by glutamate (glut, 50 µM) responsive (GCaMP6$^+$) fibers in saline (Ctrl), BOTOX (BOT, 1 U/ml), and d-tubocurarine (DTC, 25 µM) treated 3D neuromuscular co-cultures. #p<0.05 and ###p<0.001 compared with saline treated sample. In (C, D) n = 8 independent neuromuscular samples from three separate muscle patient donors. In (B–D) each symbol represents data from one muscle patient donor. Values in (B–D) are mean ±SEM. ***p<0.001.

DOI: https://doi.org/10.7554/eLife.44530.014

The following figure supplements are available for figure 3:

**Figure supplement 1.** 2D motor neuron and 3D neuromuscular co-culture functional characterization.
DOI: https://doi.org/10.7554/eLife.44530.015
**Figure supplement 2.** Evaluating functional NMJ connectivity in 2D and 3D neuromuscular co-cultures.
DOI: https://doi.org/10.7554/eLife.44530.016

decrease in calcium transient activity and an absence of tissue contraction in response to glutamate stimulation if neuromuscular co-cultures were pre-treated with BOTOX or d-tubocurarine (*Figure 3D* and *Video 7*) doses reported by others to fully block activity (*Ko et al., 2019*; *Madden et al., 2015*).

In contrast, and as expected, we observed very few functional connections when evaluating 2D neuromuscular co-cultures matured for 2 weeks and then treated with glutamate (*Figure 3—figure supplement 2A* and *Video 8*). Indeed, a prior report of 2D human neuromuscular co-cultures performed functional assays only after 60 days of culture (*Steinbeck et al., 2016*). To confirm that the muscle fibers possessed functional AChR channels, despite limited innervation at this time point, the 2D co-cultures were stimulated with ACh (100 µM). Calcium transients visualized by tracking GCaMP signal indicated the presence of ACh responsive muscle fibers in close proximity to the MN cluster (*Figure 3—figure supplement 2A* and *Video 8*).

Next, to determine the maximum length of the functional connectivity between the MN cluster and the muscle fibers in 3D cultures, we generated neuromuscular tissues using GCaMP6 transduced muscle progenitor cells and a single MN cluster. On the second week of co-culture, calcium transients arising from the 3D neuromuscular tissues were recorded during glutamate (50 µM) stimulation. Analysis of pre- and post-stimulation videos, to identify glutamate responsive fibers and substract spontaneously active fibers, indicated an average maximum functional connectivity length of

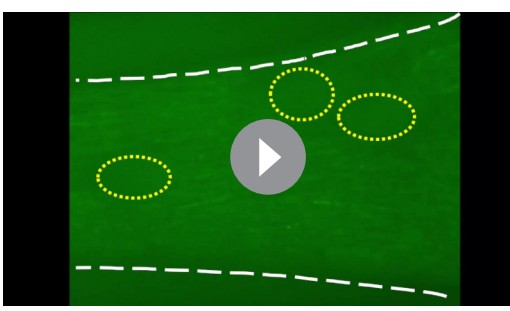

**Video 6.** Neuromuscular co-cultures are functionally innervated. A representative epifluorescence time-lapse video in which GCaMP6 transduced muscle cells co-cultured with pluripotent stem cell-derived motor neurons for 14 days in three-dimensions are first treated with HBSS saline solution, followed by L-glutamate (50 µM), and then acetylcholine (10 µM). White dashed lines outline the muscle tissue and yellow dotted circles outline motor neuron clusters.
DOI: https://doi.org/10.7554/eLife.44530.017

1042.7 ± 104.5 µm at this time-point. As expected, the number of innervated fibers decreased as the distance from the MN cluster increased (*Figure 3—figure supplement 2B*).

Finally, we performed electrophysiological recording to directly address the functional properties of the neuromuscular junctions. Using current clamp, we observed spontaneous endogenous endplate potentials (EPPs) from single muscle fibers that were proximal to the MN cluster (*Figure 3—figure supplement 2C*), which were absent in muscle-alone cultures (*Figure 1—figure supplement 3E*). Upon glutamate stimulation, the frequency of EPPs was increased (*Figure 3—figure supplement 2D*), whereas the amplitude remained unchanged (*Figure 3—figure supplement 2E*). These results support the notion that the MNs were stimulated by glutamate to release neurotransmitter into the NMJ. Moreover, in these muscle fibers, we captured an event that resembled an action potential in response to glutamate stimulation (*Figure 3—figure supplement 2F*). The event was characterized by a ~ 26 mV depolarization followed by a small plateau phase lasting ~8.5–14.5 milliseconds, but the absence of an afterhyperpolarization. Notably, we never observed spontaneously occurring action potentials.

Together, these studies indicate that 3D neuromuscular co-cultures support efficient functional innervation that occurs faster than previously reported for 2D neuromuscular co-cultures (*Steinbeck et al., 2016*).

## 3d human neuromuscular co-cultures to model adult NMJ development

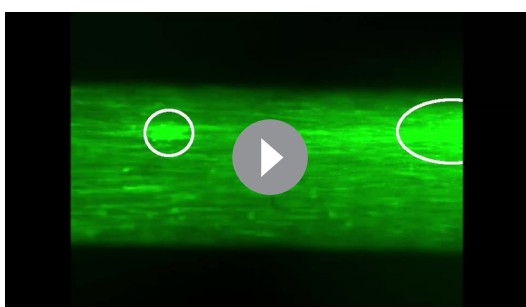

**Video 7.** Synchronous muscle fiber contractions in response to neuromuscular co-culture L-glutamate stimulation. A representative epifluorescence time-lapse video in which GCaMP6 transduced muscle cells co-cultured with GFP-expressing induced pluripotent stem cell-derived motor neurons in three-dimensions demonstrate synchronous contraction in response to treatment with L-glutamate (50 µM) at Day 14 of culture. BOTOX (1 U/ml) and d-tubocurarine (25 µM) treatments blocked the glutamate induced muscle fiber contractions. White circles outline the location of motor neuron clusters.
DOI: https://doi.org/10.7554/eLife.44530.018

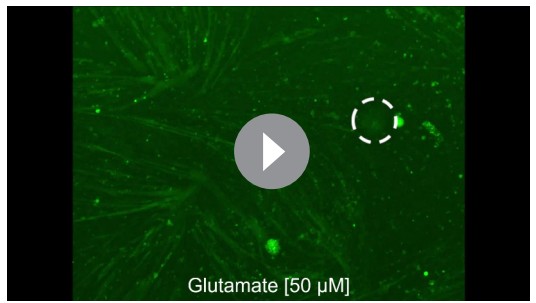

Glutamate [50 µM]

**Video 8.** 2D neuromuscular co-culture innervation and AChR development is limited. A representative epifluorescence time-lapse video in which GCaMP6 transduced muscle cells co-cultured with pluripotent stem cell-derived motor neurons for two-weeks in 2D culture are first treated with L-glutamate (50 µM) on Day 14. On Day 15 the co-culture is pre-treated with Waglerin-1 (WTX-1) and then stimulated with L-glutamate, followed by acetylcholine (10 µM). Muscle fiber calcium transients are visualized in green by following the GCaMP6 calcium reporter.
DOI: https://doi.org/10.7554/eLife.44530.019

## and disease

Next, given the high degree of innervation achieved in our neuromuscular co-cultures, we hypothesized that the 3D model might be capable of supporting the gamma (embryonic) to epsilon (adult)-subunit switch that was not observed in 2D human neuromuscular co-cultures (*Steinbeck et al., 2016*). Selective transcription of the AChR subunits occurs during different developmental stages (*Martinou et al., 1991*) and neural derived glycoprotein neuregulin-1 (NRG1), a motor neuron-derived factor, is thought to stimulate expression of the epsilon subunit of the AChR gene (CHNRE), which encodes an adult muscle AChR subunit (*Jo et al., 1995*; *Falls et al., 1993*). Using western blot experiments, we confirmed the expression of NRG1-β1 in our PSC-derived MNs (*Figure 4—figure supplement 1A*). Next, we quantified CHRNE expression in our 2D and 3D muscle-alone cultures and neuromuscular co-cultures. We observed a significant increase in the expression of the CHRNE gene in co-cultures compared to muscle-alone cultures, in both 2D and 3D, after two weeks of culture (*Figure 4—figure supplement 1B*), suggesting involvement of MN-derived trophic factors in CHRNE gene expression. To test whether the increase can be associated with NRG1-β1-mediated induction of the CHRNE gene, we supplemented our 2D and 3D muscle-alone cultures with recombinant NRG1- β1 (5 nM) and detected a significant increase in CHRNE expression in the supplemented muscle fiber cultures (*Figure 4—figure supplement 1B*). Treating 3D muscle-alone cultures with motor neuron-derived conditioned media did not induce epsilon subunit gene expression above untreated 3D muscle alone cultures (CHRNE 0.5 ± 0.4% of GAPDH expression), suggesting that MN axon contact with muscle fibers may be necessary to locally deliver concentrated neurotrophic factors and modulate epsilon gene expression in muscle fibers. Further, given the limited innervation observed in 2D co-cultures at this time-point (*Figure 3—figure supplement 2A* and *Video 8*), we speculate that an NMJ-independent mechanism of localized neurotrophic factor delivery contributes to CHRNE gene expression in muscle cells.

We next evaluated AChR epsilon expression at the protein level and found that it was upregulated in 3D co-cultures, but not in 2D co-cultures (*Figure 4A–B*). The upregulation of AChR epsilon protein expression in 3D co-cultures was accompanied by a significant increase in AChR beta and no change in the AChR gamma subunit (*Figure 4A–B*), in support of studies concluding that gamma subunit transcription and translation does not appear to influence the onset or magnitude of epsilon expression (*Witzemann et al., 1996*; *Yampolsky et al., 2008*), and hinting that some embryonic AChRs may remain. MN-dependent changes in AChR subunit protein levels (beta, gamma, and epsilon) were not observed in 2D co-cultures (*Figure 4A*). These observations support the notion that AChR epsilon protein stability is influenced by the degree of muscle fiber and NMJ activity (*Caroni et al., 1993*; *Missias et al., 1996*; *Witzemann et al., 2013*; *Xu and Salpeter, 1997*; *Yampolsky et al., 2008*).

We then sought to determine if the 3D human neuromuscular co-culture system was suitable for modeling congenital myasthenic syndromes caused by mutations in CHRNE by blocking the AChR-epsilon subunit using Waglerin-1 (WTX); a peptide that selectively binds and blocks the epsilon subunit of the muscle AChR (*McArdle et al., 1999*). The AChR channel contains two binding sites for ACh, and one of those sites sits between the epsilon and a beta subunit in the adult AChR. Thus, if the epsilon subunit is functionally integrated into the AChR in neuromuscular co-cultures, then WTX treatment is expected to dampen calcium transients following glutamate stimulation by decreasing the statistical likelihood that the AChR channel will open (*Jha and Auerbach, 2010*; *Ohno et al., 1996*). In these experiments, 3D neuromuscular tissues were generated using GCaMP6 transduced muscle progenitor cells and each tissue was stimulated with glutamate twice: pre- and post WTX treatment (1 μM), with a 24 hr recovery time allocated between each stimulation. We recorded videos during glutamate stimulation and then quantified the maximal tissue area containing glutamate responsive fibers by analyzing the GCaMP6 fluorescence signal in the same tissue pre- and post-WTX treatment at defined regions of interests (*Figure 4C* and *Video 9*). Consistently, we observed a 46.47 ± 15% (N = 3; p<0.05) decrease in glutamate responsive tissue area following glutamate stimulation in WTX pre-treated neuromuscular tissues (*Figure 4C*). Similar results were obtained by analyzing calcium transients in individual fibers pre- and post WTX treatment in response to glutamate stimulation (*Figure 4D*; 55.8 ± 1.8% decrease). This analysis also revealed a subset of WTX-treatment refractory single fibers (*Figure 4C*, white arrowheads), indicating that not all AChRs in the 3D neuromuscular co-culture undergo the developmental switch by this time point. We performed similar

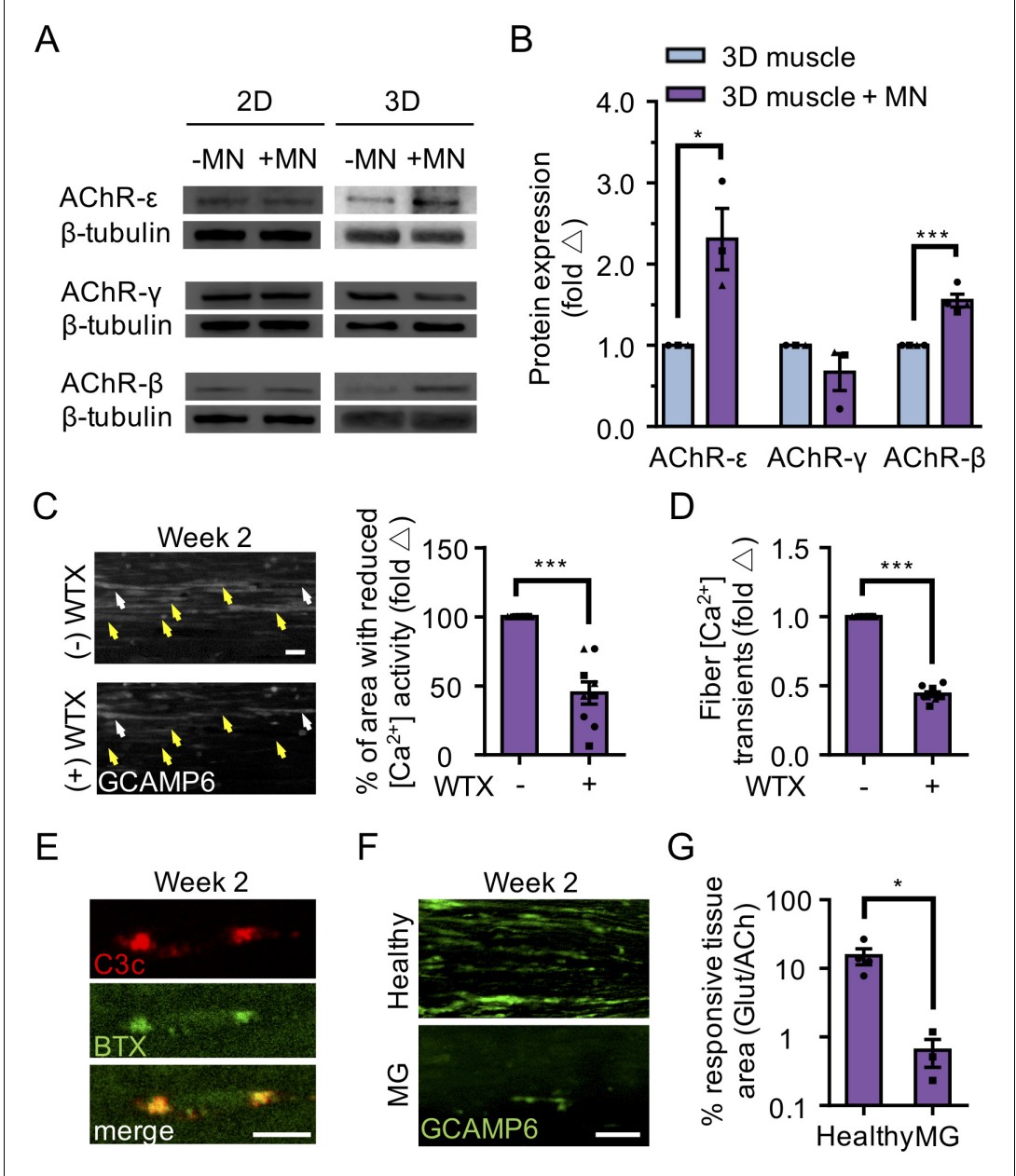

**Figure 4.** 3D neuromuscular co-cultures enable disease modeling of adult NMJ in vitro. (**A**) Representative western blot images of nicotinic acetylcholine receptor subunit epsilon (nAChR-ε), gamma (nAChR-γ), and beta (nAChR-β) proteins in 2D and 3D muscle-alone (-MN) and neuromuscular co-cultures (+MN) at two weeks of culture. (**B**) Bar graph quantification of nACHR subunit ε, γ, and β protein expression in 3D muscle (blue) and 3D neuromuscular (purple) cultures. Values are normalized to 3D muscle cultures. (**C**) (left panel) Representative epifluorescence images of GCAMP6 signals in response to glutamate (glut) stimulation before (top panel) and after (bottom panel) 3D neuromuscular co-culture treatment with Waglerin 1 (WTX-1). Yellow arrowheads point out fibers with dampened GCAMP6 fluorescence signal following WTX-1 treatment. White arrowheads indicate fibers that did not dampen calcium handling after WTX-1 treatment. Scale bar, 50 μm. (right panel) Bar graph indicating the percentage of 3D neuromuscular co-culture tissue area occupied by glutamate responsive fibers (GCaMP6+) before (-) and after (+) WTX (1 μM) treatment. (**D**) Bar graph quantifying glutamate-induced GCAMP6 signals from individual fibers before (-) and after (+) WTX-1 treatment. In (**C–D**), data is normalized to (-) WTX condition. For (**B–D**), n = 9 independent muscle or neuromuscular samples from three muscle patient donors. A minimum of 50 fibers were analyzed for data presented in (**D**). (**E**) Representative confocal images of a 3D muscle culture co-treated with Myasthenia gravis (MG) patient IgG and human complement and then immunostained for human complement component C3c (red, top) and α-bungarotoxin (BTX, green, middle). Bottom panel is a merged image of the top and middle panels. Scale bars, 10 μm. (**F**) Representative epifluorescence images of GCaMP6 signals from a glutamate stimulated 3D neuromuscular co-culture following a 72 hr treatment with 300 nM of healthy (top panel) or MG (bottom panel) patient IgG and human complement. Scale bars, 100 μm. (**G**) Bar graph indicating the percent tissue area occupied by glutamate (glut, 50 μM) responsive (GCaMP6+) fibers in

*Figure 4 continued on next page*

*Figure 4 continued*

healthy and MG patient IgG treated 3D neuromuscular co-cultures. Data normalized to the total area of ACh responsive (GCaMP6$^+$) tissue in each co-culture. n = 4 independent neuromuscular tissues treated with healthy IgG and three neuromuscular tissues each treated with serum IgG from one of three separate MG patient donors. In (**B–D**) and (**G**) each symbol represents data from one patient donor. Values in (**B–D**) and (**G**) are mean ±SEM. *p<0.05 ***p<0.001.
DOI: https://doi.org/10.7554/eLife.44530.020
The following figure supplement is available for figure 4:

**Figure supplement 1.** NMJ development and disease modeling studies using 3D neuromuscular co-cultures.
DOI: https://doi.org/10.7554/eLife.44530.021

experiments on 2D neuromuscular co-cultures (*Video 8*), and 3D muscle-alone cultures (*Video 10*), but did not observe calcium transient changes. Importantly, 3D neuromuscular tissue GCaMP signal was not dampened by serial glutamate stimulation (see methods) excluding the possibility that GCaMP dampening was the result of glutamate neurotoxicity.

Collectively, this data suggests that the 3D neuromuscular co-culture platform allows for rapid and easy modeling and study of diseases impacting the adult human NMJ.

## 3d human neuromuscular co-cultures to model myasthenia gravis

To demonstrate the tractability and robustness of the 3D neuromuscular co-culture system to study human disease, we treated co-culture tissues with IgG isolated from three patients afflicted with AChR-targeted myasthenia gravis (*Table 1*) to model autoimmune myasthenia gravis. Myasthenia gravis (MG) is an autoimmune disease manifesting as muscle weakness caused by the production of autoantibodies that alter, block, or destroy NMJ receptors required for signal transmission. IgG and complement deposit at the NMJ eliciting inflammation and subsequent destruction of AChRs on the postsynaptic NMJ membrane (*Engel et al., 1977*). Therefore, we treated our neuromuscular tissues with IgG (300 nM) isolated from healthy or MG patients together with human serum, which contains complement. Localized deposition of complement on BTX stained AChRs was confirmed by staining for the complement C3c protein one-day after co-treating muscle tissues with MG IgG and active human complement (*Figure 4E*). We recorded neuromuscular co-culture GCaMP signals arising from L-glutamate (50 µM) stimulation after a 3 day incubation with healthy or MG IgG (*Figure 4F*), to visualize NMJ activity. We then stimulated the co-cultures with ACh to quantify the total area occupied by muscle fibers. Our analysis revealed a clear decrease in the area of stimuli responsive muscle fibers (*Figure 4G* and *Videos 11–12*) and a decline in the area of the tissue responsive to ACh stimulation (*Figure 4—figure supplement 1C–D* and *Videos 11–12*) when tissues were treated with MG compared to healthy patient IgG.

This study demonstrates the simplicity of implementing the 3D neuromuscular co-culture system to the application of modeling a human NMJ disorder in culture.

## Discussion

Here we report a simple method to co-culture 3D human skeletal muscle fiber tissues together with human PSC-derived MNs, while performing comparative studies to uncover biological inquiries and processes that are enabled by the availability of a 3D human neuromuscular culture system. Our focus is distinct, but complementary to recent work reporting strategies to co-culture hPSC-derived skeletal muscle progenitors and motor neurons in 3D (*Maffioletti et al., 2018*; *Osaki et al., 2018*), and that implement the 3D co-culture approach in the context of a microfluidics platform to model human disease (i.e. ALS) (*Osaki et al., 2018*). We demonstrate that functional innervation is achieved in 3D, but not 2D neuromuscular co-cultures, within 2 weeks of culture. Indeed, we find that innervation in 3D neuromuscular co-cultures is ~4 fold faster and more efficient than a prior report of a 2D human neuromuscular co-culture system (*Steinbeck et al., 2016*), and we show that this simplifies and expedites studies of myasthenia gravis in a dish. With side-by-side comparisons of 2D and 3D muscle-alone and neuromuscular co-cultures, we confirmed that CHRNE transcription is supported by MN co-culture in 2D and 3D, and then show that the AChR epsilon subunit protein is only functionally integrated into the AChR in the context of 3D neuromuscular co-cultures. Therefore, this is

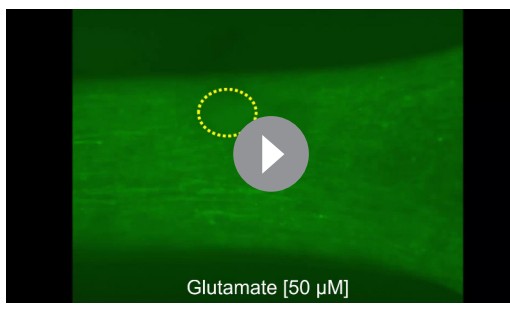

Glutamate [50 µM]

**Video 9.** 3D neuromuscular co-cultures enable studies of the AChR epsilon subunit. A representative epifluorescence time-lapse video in which GCaMP6 transduced muscle cells co-cultured with pluripotent stem cell-derived motor neurons for two-weeks in 3D culture are treated with L-glutamate (50 µM) on Day 14, and then with Waglerin-1 (WTX) followed by L-glutamate (50 µM) on Day 15. Muscle fiber calcium transients are visualized in green by following the GCaMP6 calcium reporter. A yellow dotted line outlines the location of the motor neuron cluster.
DOI: https://doi.org/10.7554/eLife.44530.022

the first report of a culture method to study the de novo AChR gamma to epsilon subunit developmental switch in culture and to model diseases of the adult human NMJ in a dish.

Our side-by-side comparison of human skeletal muscle fiber cultures in 2D and 3D indicates the structural and functional advantages of a 3D culture model over currently available 2D systems. The value of 3D culture is reported in previous studies for other organs (*Lancaster and Knoblich, 2014*), and in this study we provide the first quantitative evidence that 3D culture conditions lend to the maturation of multinucleated muscle fibers due to their capability to accommodate the inherent contractile nature of the muscle fibers in the long-term. This in turn leads to muscle fiber hypertrophy, improved calcium handling, and muscle fiber maturation as evidenced by expression of adult forms of MHC, and elaborated clustering of AChRs. This makes 3D neuromuscular cultures an ideal platform for studying NMJ synaptogenesis given the inherently long process required for functional NMJ development to occur. However, it should be noted that single fiber level analyses may be easier to perform in a 2D culture setting. As such, we expect that focusing efforts on modifying 2D cultures to control the microenvironment in ways that can accommodate myofiber contractility and alignment, might result in the formation of functional NMJs in vitro at earlier time-points, as was recently demonstrated in work with rodent myoblasts and neural cells (*Ko et al., 2019*). If successful, it is feasible to imagine de novo AChR subunit switching in 2D neuromuscular cultures as well.

Consistently, electrophysiological recordings of single muscle fibers in these neuromuscular co-cultures detected putative endogenous and glutamate-stimulated EPPs, suggesting that MNs form functional neuromuscular junctions, similar to that observed in in vivo mammalian models. However,

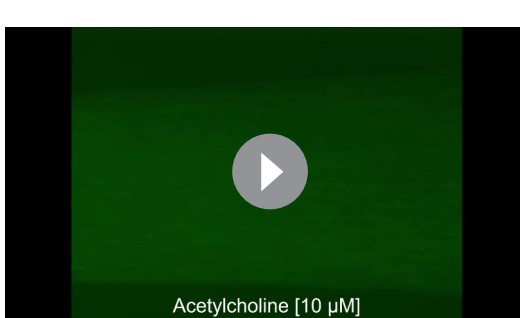

Acetylcholine [10 µM]

**Video 10.** Waglerin-1 treatment does not dampen ACh induced muscle fiber calcium transients in 3D muscle alone cultures. A representative epifluorescence time-lapse video in which GCaMP6 transduced muscle cells cultured two-weeks in 3D culture are first treated with ACh (10 µM) on Day 14, and then pre-treated with Waglerin-1 (WTX, 1 µM) followed by another ACh (10 µM) stimulation. Muscle fiber calcium transients are visualized in green by following the GCaMP6 calcium reporter.
DOI: https://doi.org/10.7554/eLife.44530.023

we will note that in the absence of pre- or post-synaptic blocking studies during these recordings, we can speculate, but not conclude definitively, that the activity we recorded was in fact evoked by glutamate stimulation. The properties of action potentials recorded in these cultures is consistent with the possibility that the 3D human neuromuscular co-cultures are not fully mature. This observation concurs with the relatively small muscle fibers, and indicates that additional chemical or physical cues are necessary to mature the tissues further. We saw a large action potential-like response in 1 out of the seven fibers we assessed. This suggests that although the NMJs are functional, exhibit endogenous activity, and that motor neurons respond to glutamate application by increasing the basal rate of neurotransmitter release, most motor neurons are still in an immature state and do not trigger synchronous neurotransmitter release in response to glutamate application. This could be at the level of action potential generation in response to glutamate application, or converting

**Table 1.** Myasthenia Gravis patient information.

| Patient ID | Sex | Anti-AChR titer (nM) |
|---|---|---|
| MG#1 | Male | >10 |
| MG#2 | Female | 8.6 |
| MG#3 | Female | >10 |

DOI: https://doi.org/10.7554/eLife.44530.024

action potentials to synchronous release at the pre-synapse. We anticipate increasing culture time, providing electrical stimulation, and/or adding trophic or synaptogenesis factors might improve the maturity of the neuromuscular co-culture and their connections.

Alternatively, our electrophysiological conclusions may simply reflect the technical challenges we faced in the course of our recordings (see Materials and Methods). In general, analyses at the single fiber level require the user to develop or implement tools or adapt the protocol to improve feasibility. Electrophysiological recordings in the co-culture system are feasible, but highly challenging for a number of reasons. First, spontaneous or induced muscle tissue contractions in 3D neuromuscular co-cultures frequently resulted in the loss of pipette contact with the cell membrane during recordings. As a result, recording where glutamate is added to the culture bath as a stimulation method, which elicits tissue movement via multiple muscle fibers contracting in unison (as seen in *Video 7*), were challenging and as such, only successful in few events, as we reported in the manuscript. However, this challenge can be overcome by performing targeted stimulation of single motor neurons with electrical or neurotransmitter stimulation, or by stimulating with blue light in the case of motor neurons genetically modified to express a light sensitive channel (e.g.) channelrhodopsin. In addition, identifying innervated muscle fibers is challenging. Using fluorescently labeled muscle cells (e.g. GCaMP6+, membrane anchored fluorophore) and motor neuron (e.g. HB9-GFP, mCherry, neurofilament-GFP) cells dramatically improves the success rate in identifying innervated muscle fibers in 3D neuromuscular co-cultures. Notably, the 3D nature of the co-culture system reduces the incidence of pipette breakage common in studies of plastic cultured myotubes. However, without added myotube maturation through contraction regimes or otherwise, it should be noted that myotubes in cul-

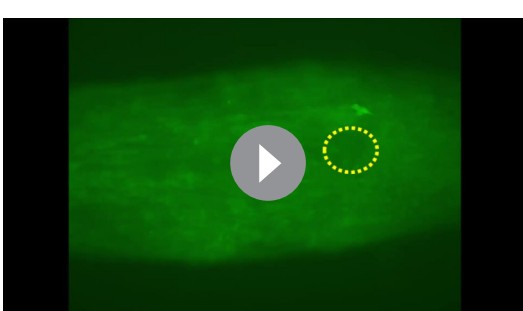

**Video 11.** The influence of Myasthenia gravis autoantibodies on NMJ activity is easily studied in 3D neuromuscular co-cultures. A representative epifluorescence time-lapse video in which GCaMP6 transduced muscle cells co-cultured with pluripotent stem cell-derived motor neurons for 14 days in three-dimensions are first stimulated with L-glutamate (50 µM) to assess neuromuscular junction transmission, and then with acetylcholine (100 µM) to visualize all fibers in the culture. These cultures were treated for 3 days (Day 11 to Day 14) with Myasthenia gravis patient IgG (300 nM) and 2% human serum. A yellow dotted line outlines the location of the motor neuron cluster.

DOI: https://doi.org/10.7554/eLife.44530.025

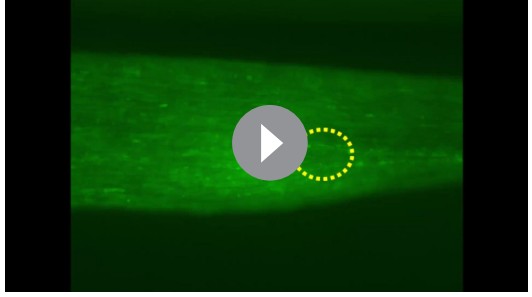

**Video 12.** 3D neuromuscular co-cultures treated with healthy patient IgG and complement display normal calcium transients in response to glutamate stimulation. A representative epifluorescence time-lapse video in which GCaMP6 transduced muscle cells co-cultured with pluripotent stem cell-derived motor neurons for 14 days in three-dimensions are stimulated first with L-glutamate (50 µM) to assess neuromuscular junction transmission, and then with acetylcholine (100 µM) to visualize all fibers in the culture. These cultures were treated for 3 days (Day 11 to Day 14) with healthy patient IgG (300 nM) and 2% human serum. A yellow dotted line outlines the location of the motor neuron cluster.

DOI: https://doi.org/10.7554/eLife.44530.026

ture are somewhat smaller than those in adult animals, which can introduce some difficulty in recording.

Perhaps most strikingly, 3D neuromuscular cultures possess AChRs containing functional adult AChR epsilon subunit, which is, to our knowledge, the first report of a system that supports the de novo gamma to epsilon AChR subunit switch in culture. Given challenges associated with maturing hPSC-derived skeletal muscle fibers beyond embryonic-like states, we hypothesize that our success may be due in part to the use of primary adult human myoblasts. In a proof-of-concept study, we demonstrate the application of our NMJ model to study adult NMJ activity by using a peptide that specifically blocks the epsilon subunit. Treatment with the peptide dampened glutamate-induced GCaMP6 calcium reporter activity in neuromuscular co-cultures demonstrating the utility of the system for adult NMJ studies.

Tissue culture affords the opportunity to deconstruct the complexity of a tissue system and to systematically rebuild complexity as a means to identify physical and chemical factors that influence biological processes. This method is particularly powerful in studies of the NMJ where decoupling nerve and muscle influences during development and in the adult, within the context of an animal model, is confounded by tissue death. Through an iterative comparison of 2D and 3D muscle alone and neuromuscular co-cultures, we found that CHRNE transcript expression is upregulated in both 2D and 3D neuromuscular co-cultures. Bathing 2D or 3D muscle fiber cultures with a high concentration of recombinant neuregulin-1 phenocopied the effect of MN co-culture on CHRNE transcript, but CHRNE transcript levels were not induced in 3D muscle-alone culture treated with conditioned media from MNs. Since our PSC-derived motor neurons express neuregulin-1 protein, but we do not observe appreciable NMJ activity in our 2D neuromuscular co-cultures, we speculate that if CHRNE transcript induction is reliant on NRG-1, then localized MN-mediated delivery of the protein may be necessary to achieve physiologically relevant concentrations of the protein, and that transmission via the NMJ is not required. Importantly, epsilon protein levels further increased and its function was detected (WTX-responsivity) only in the context of 3D neuromuscular co-cultures.

Our culture data indicates that the epsilon subunit of the AChR is subjected to post-transcriptional modifications and/or intracellular trafficking events that are only supported in the context of 3D neuromuscular co-culture. Indeed, our observations that muscle fibers established in 3D culture are more mature (*Figure 1* and *Figure 1—figure supplement 1* and **2**) and that 3D neuromuscular co-cultures exhibit spontaneous endogenous endplate potentials (*Figure 3* and *Video 7*) fit well with studies linking muscle fiber maturation state and activity to AChR subunit conversion and stability (*Caroni et al., 1993*; *Missias et al., 1996*; *Witzemann et al., 2013*; *Xu and Salpeter, 1997*; *Yampolsky et al., 2008*). Through the availability of a methodology supporting de novo adult NMJ development, it is now possible to delve deeper into the mechanisms regulating metabolic stability of the epsilon subunit in normal development and in disease states. Studies aimed at understanding the intricacies of subunit integration, recycling, and stability are poised for exploration upon the availability of antibodies that allow for immunostaining studies of the human epsilon protein, or the generation of genetically modified lines in which subunits are fluorescently tagged.

In summary, this approach to model the adult human NMJ in a dish provides a versatile and simple way to study skeletal muscle and NMJ development, but more importantly, constitutes the first report of a method to study adult, rather than embryonic, human NMJ activity in as early as two weeks of co-culture time. Our calcium reporter neuromuscular tissues can easily be integrated with other optogenetic methods (*Steinbeck et al., 2016*), and would benefit from such an approach, to further elucidate synaptic transmission mechanisms of adult NMJ, such as adult AChR conductance. Furthermore, neuromuscular co-cultures may be integrated with other neuron populations such as upper MNs and/or myelinating Schwann cells to support studies aimed at a better understanding of signal transmission in the central nervous system. Finally, our method is amenable to modeling diseases that target the adult NMJ (e.g. congenital myasthenia gravis, Duchenne muscular dystrophy (*Xu and Salpeter, 1997*) and to assess drugs to support personalized medicine applications.

## Materials and methods

### Human primary myoblast derivation and propagation

Small skeletal muscle samples (~1 cm$^3$) were obtained from the multifidus muscle of patients undergoing lumbar spine surgery. Primary myoblast and fibroblast-like cell lines were established and maintained as previously described (*Blau and Webster, 1981*). Briefly, human skeletal muscle samples were minced and then dissociated into a single cell slurry with clostridium histolyticum collagenase (Sigma, 630 U/mL) and dispase (Roche, 0.03 U/mL) in Dulbecco's Modified Eagle's medium (DMEM; Gibco). The cell suspension was passed multiple times through a 20 G needle to facilitate the release of the mononucleated cell population and subsequently depleted of red blood cells with a brief incubation in red blood cell lysis buffer (*Table 2*). The resulting cell suspension containing a mixed population of myoblasts and fibroblast-like cells was plated in a collagen-coated tissue culture dish containing myoblast growth medium: F-10 media (Life Technologies), 20% fetal bovine serum (Gibco), 5 ng/mL basic fibroblast growth factor (bFGF; ImmunoTools) and 1% penicillin-streptomycin (Life Technologies). After one passage, the cell culture mixture was stained with an antibody recognizing the neural cell adhesion molecule (NCAM/CD56; BD Pharmingen; *Table 3*), and the myogenic progenitor (CD56$^+$) and fibroblast-like cell (CD56$^-$) populations were separated and purified using fluorescence-activated cell sorting (FACS) and maintained on collagen coated dishes in growth medium. Subsequent experiments utilized low passage cultures (P4—P9).

### Human primary myoblast two-dimensional culture

Primary human myoblasts were mixed with primary human muscle fibroblast-like cells at the following ratios: CD56$^+$ (95%) and CD56$^-$ (5%). For Geltrex culture dish coating, 1 mg of Geltrex was resuspended in 12 mL of ice-cold DMEM and 1 mL was transferred to each well of a 12 well plate. Plates were incubated at 37°C overnight. DMEM was aspirated the next day just prior to cell culture. $3 \times 10^6$ cells resuspended in bFGF-free myoblast growth media (*Table 2*) were plated into each Geltrex (Life Technologies) coated well. The growth media was exchanged 2 days later with myoblast differentiation medium (*Table 2*). Half of the culture media was exchanged every other day thereafter. In some experiments (*Figure 1—figure supplement 2G-H*), fibrinogen was supplemented into the differentiation media at 10 µg/mL to control for the effect of fibrinogen receptor ligation on two-dimensional (2D) muscle fiber differentiation.

**Table 2.** Cell Culture Media and Solutions

| # | Name | Details |
|---|------|---------|
| 1 | Blocking solution | 20% goat serum, 0.3% Triton-X 100 in PBS |
| 2 | Fibrinogen stock solution | 10 mg / mL fibrinogen in 0.9% (wt/v) NaCl solution in water |
| 3 | Human fibroblast growth media | Dulbecco's Modified Eagle's medium (DMEM), 10% fetal bovine serum, 1% penicillin-streptomycin |
| 4 | Human myoblast differentiation media | Dulbecco's Modified Eagle's medium (DMEM), 2% horse serum, 10 µg / mL insulin, 1% penicillin-streptomycin |
| 5 | Human myoblast growth media | Ham's F-10 nutrient mix, 20% fetal bovine serum, 5 ng / mL basic fibroblast growth factor, 1% penicillin-streptomycin |
| 6 | Hydrogel mixture | Dulbecco's Modified Eagle's medium (DMEM), 4 mg / mL bovine fibrinogen, Geltrex (20% v / v), thrombin (0.2 unit/mg fibrinogen) |
| 7 | Milk based blocking solution | 5% (wt/v) skim milk (BioShop) in TBST |
| 8 | Red blood cell lysis buffer | 15.5 mM NH4Cl, 1 mM KHCO3, 10 µM EDTA |
| 9 | Tris-buffered saline Tween (TBST) | 50 mM Tris (BioShop), 150 mM NaCl (Sigma), 0.1% (v/v) Tween 20 (BioShop) |

DOI: https://doi.org/10.7554/eLife.44530.027

**Table 3.** List of primary antibodies.

| # | Antibody | Species | Dilution | Source |
|---|---|---|---|---|
| 1 | Alexa Fluor 647 mouse anti-human CD56 | Mouse | 1:20 | BD Pharmingen |
| 2 | Anti-200 kD neurofilament heavy (SMI-32) | Rabbit | 1:200 | Abcam |
| 3 | Anti-C3c (FITC) | Rabbit | 1:200 | Abcam |
| 4 | Anti-HB9/HLXB9 | Rabbit | 1:100 | Abcam |
| 5 | Anti-vimentin | Rabbit | 1:100 | Abcam |
| 6 | Anti- β -tubulin | Rabbit | 1:5000 | Cell Signaling |
| 7 | DRAQ5 | - | 1:1000 | ThermoFisher |
| 8 | Hoechst 33342 | - | 1:1000 | ThermoFisher |
| 9 | Islet-1 | Goat | 5 µg/ml | R and D systems |
| 10 | NRG1- β1 | Mouse | 1:500 | R and D Systems |
| 11 | Monoclonal anti-β-actin-peroxidase | Mouse | 1:50000 | Sigma |
| 12 | Monoclonal mouse anti-human desmin | Mouse | 1:100 | Dako |
| 13 | MuSK (PA5-14703) [WB] | Rabbit | 1:1000 | Invitrogen |
|  | MuSK (PA1-1741) [IF] | Rabbit | 1:50 | Invitrogen |
| 14 | Myosin heavy chain - embryonic | Mouse | 1:50 | DSHB |
| 15 | Myosin heavy chain - fast | Mouse | 1:50 | DSHB |
| 16 | Myosin heavy chain - slow | Mouse | 1:50 | DSHB |
| 17 | Myosin heavy chain - pan | Mouse | 1:50 | DSHB |
| 18 | Nicotinic acetylcholine receptor β | Rabbit | 1:2000 | Novus |
| 19 | Nicotinic acetylcholine receptor epsilon | Rabbit | 1:1000 | Novus |
| 20 | Rapsyn | Mouse | 1:1000 | Abcam |
| 21 | Sarcomeric alpha-actinin | Mouse | 1:200 | Sigma |
| 22 | α-Bungarotoxin, Alexa Fluor 647 conjugate | - | 1:500 | ThermoFisher |

DOI: https://doi.org/10.7554/eLife.44530.028

## PDMS mold fabrication for 3d human muscle tissue culture

Standard 12-well culture plates were coated with 500 µL of liquid PDMS (184 Silicone Elastomer Kit, 10 parts elastomer to one part curing). After curing at 50℃ for at least 3 hr, another 750 µl of liquid PDMS was added to each well and a laser cut, dumbbell shaped piece of acrylic (middle channel dimensions = 14 mm by 2.75 mm; side chamber dimensions = 5.7 mm by 2.5 mm) was submerged in the liquid PDMS. Plates were then placed within a vacuum chamber for a minimum of 10 min to remove bubbles from the liquid PDMS. The PDMS was cured by incubating the plates in a 50℃ oven for 3 hr. Acrylic pieces were then removed from the PDMS, leaving a dumbbell-shaped depression in the PDMS, and two pieces of Velcro fabric were affixed at each end of the channel using liquid PDMS as glue (*Figure 1—figure supplement 1C*). Each well was sterilized with 70% ethanol at room temperature in a tissue culture hood for at least 30 min. At this point, plates were parafilm sealed, and stored at room temperature. Prior to use, PDMS mold wells were incubated with a 5% pluronic acid (Sigma) solution in ddH$_2$O for 12 hr at 4℃. Pluronic acid solution was aspirated and molds were rinsed with a PBS solution before seeding human muscle tissues.

Remark: The integrity of nylon hooks, as anchor points, on the Velcro pieces is critical for the successful culture of 3D tissues and preventing their immature rupture. As such, we recommend careful inspection of the Velcro pieces before to ensure latch and hooks are without defects, and after to ensure pieces are well adhered in the dish. These are critical steps pre-tissue seeding to ensure successful tissue remodeling and culture.

## Human myoblast three-dimensional culture

Three-dimensional (3D) human skeletal muscle tissues were generated in culture as previously described (*Madden et al., 2015*) with the following modification: FACS-purified CD56$^+$ myoblasts (95%) and CD56$^-$ fibroblast-like cells (5%) were incorporated into tissues. Briefly, cells at these defined ratios were resuspended in the hydrogel mixture (*Table 2*) in the absence of thrombin. Thrombin (Sigma) was added at 0.2 unit per mg of fibrinogen just prior to evenly seeding the cell/hydrogel suspension in the long channel of the dumbbell-shaped molds. Tissues were then incubated for 5 min at 37°C to expedite fibrin polymerization. Myoblast growth media (*Table 2*) lacking bFGF, but containing 1.5 mg/mL 6-aminocaproic acid (ACA; Sigma), was added. 2 days later the growth media was exchanged to myoblast differentiation medium (*Table 2*) containing 2 mg/mL ACA. Half of the culture media was exchanged every other day thereafter. In agrin treatment experiments, recombinant rat agrin (R and D Systems) was supplemented in the culture media at 50 ng/ml. In experiments using neuregulin1-β1 treatment, recombinant human neuregulin1-β1 (R and D Systems) was supplemented in the culture media at 5 nM. In both cases (agrin, neuregulin), treatment began when tissues were switched to differentiation medium and recombinant proteins were added to exchange media at 2-fold concentration. hESC differentiation to post-mitotic motor neurons.

Motor neurons were specified from WA09 hESCs (passage 25–45; WiCell) as previously described (*Lippmann et al., 2014*; *Lippmann et al., 2015*). Briefly, hESCs were maintained on Matrigel (BD Biosciences) in E8 medium with insulin added at a concentration of 2 mg / L. For differentiation, hESCs were dissociated with accutase (Life Technologies) and reseeded at $1 \times 10^5$ cells / cm$^2$ in E8 medium containing 10 μM ROCK inhibitor (Y27632; R and D Systems) in 6-well polystyrene tissue culture plates coated with 100 μg / mL poly-L-ornithine (PLO; Sigma) and 8 μg / well VTN-NC (gift from Dr. James Thomson). hESC were differentiated to OLIG2$^+$ progenitors in E6 medium containing the same insulin concentration as in E8 medium as previously described (*Lippmann et al., 2015*). For differentiation of the OLIG2$^+$ progenitors to motor neurons, cells were sub-cultured by en bloc passage, reseeded at a 1:200 ratio in Geltrex-coated 6-well plates, and differentiated for 14 days in E6 medium containing 1 μM retinoic acid (Sigma), 100 nM purmorphamine (Tocris), and 100 ng/mL sonic hedgehog (R and D systems). To push neuronal maturation before myoblast co-culture, 5 μM DAPT (Tocris) was added from days 8–14.

For a subset of experiments (*Figures 2L* and *3C–D*, *Figure 2—figure supplement 2C-E* and *Video 7*), OLIG2$^+$ progenitor cells were specified following a previously described method (*Du et al., 2015*) using GFP-expressing iPSCs (*Nazareth et al., 2013*). Next, OLIG2$^+$ progenitors were differentiated to post mitotic motor neurons following the same differentiation protocol mentioned above.

Remark: Batch-to-batch variability of small molecules and growth factors used in the motor neuron differentiation process can affect the success of the differentiation. As such, we recommend testing each new batch.

Remark: If starting with frozen vials of PSC-derived Olig2$^+$ precursor cells, note that excessive shearing of the Olig2$^+$ cells post-thaw dramatically reduces the differentiation efficiency. As such, the cell pellet post thaw should be transferred from the conical tube to the culture plate with passage through a pipette no more than three times.

## Two- and three-dimensional neuromuscular co-culture

24 hr after seeding myogenic progenitor cells for culture in 2D (as described above), 5 ESC-derived motor neuron clusters were detached and transferred to the muscle cell culture plates using a 1 ml pipette tip in myoblast media lacking bFGF, but now containing 10 ng / ml brain derived neurotrophic factor (BDNF) and 10 ng / ml glial cell line derived neurotrophic factor (GDNF). Mid-sized clusters (150 to 300 μm in diameter) were visually identified and selected for transfer. 24 hr later the media was removed and replaced with myogenic differentiation media (*Table 2*) supplemented with 10 ng / mL BDNF and 10 ng / mL GDNF. Half of the culture media was exchanged every other day thereafter and included both neurotrophic factors at 2-fold concentration.

For 3D neuromuscular tissues, 3D skeletal muscle tissue cell/hydrogel suspension was prepared as described above. Motor neuron clusters were transferred manually to the cell/hydrogel suspension at a ratio of 5 clusters per tissue. Thrombin was added and tissues were seeded into dumbbell-shaped molds as described above. Myoblast growth media lacking bFGF, but containing 2 mg / mL

6-aminocaproic acid (ACA; Sigma), 10 ng / mL BDNF, and GDNF was added to tissues. Two days later the culture media was exchanged to myogenic differentiation media (*Table 2*) and supplemented with 10 ng / mL BDNF and 10 ng / mL GDNF. Half of the culture media was exchanged every other day thereafter and included both neurotrophic factors at 2-fold concentration. 2D and 3D muscle-alone cultures serving as neuromuscular co-culture controls were also supplemented with BDNF and GDNF. Co-cultures were analyzed at time points indicated in the figures and legends.

For a subset of experiments (, *Figure 2—figure supplement 2F*, *Figure 3—figure supplement 2A*, *Figure 4—figure supplement 1C–D*, and *Videos 11–12*), immortalized myogenic progenitor cells (AB1167, from fascia lata muscle of a healthy 20 year old male), were employed. Briefly, human-derived skeletal muscle cell (hSMC) lines used in this work were derived from healthy subjects and were then immortalized by transduction with human telomerase-expressing and cyclin-dependent kinase 4-expressing vectors, as previously described (*Mamchaoui et al., 2011*). For these experiments we transduced the immortalized human myogenic progenitor cells with lentiviral particles to express GCaMP6, a fluorescent calcium indicator (AddGene plasmid #65042; Trono Lab packaging and envelope plasmids, Addgene plasmid #12260 and 12259), as described on the Broad Institute website (https://portals.broadinstitute.org/gpp/public/resources/protocols). The cell population was then sorted for GFP expression to enrich transduced cells and were then further expanded in myoblast growth media (*Table 2*). Methods to produce 3D neuromuscular tissues using immortalized myogenic progenitor cells were exactly as those described above for primary human muscle progenitors with the exception that fibroblast-like cells were excluded.

Remark: Collection of motor neuron clusters that are ~300 µm in diameter for muscle-motor neuron co-cultures is performed using a tip of a fine precision curved-tip forceps using an inverted microscope with 4X magnification objective. Neurites are dissected using the forceps and the released motor neuron cluster is carefully transferred to a 1.5 ml microcentrifuge tube containing E6 medium. Attention should be given to visually inspect and avoid collection of undifferentiated cells in this step. Qualitatively, motor neuron clusters possessing a densely packed cluster of nuclei yielded reproducible results.

Remark: High levels of spontaneous contractile activity of myofibers is observed in 3D muscle-motor neuron co-cultures post day 10 differentiation which might lead to their premature rupture at an earlier time before day 14. As such, 3D co-culture tissues should be inspected carefully post-day 10 differentiation.

## Immunostaining and fluorescence microscopy

2D cultures and 3D tissue whole mounts were fixed in 4% PFA for 10 min and then washed with phosphate buffered saline (PBS). Following fixation, samples were incubated in blocking solution (*Table 2*) for at least 1 hr. Samples were incubated in primary antibody solutions (*Table 3*) diluted in blocking solution (*Table 2*) overnight at 4°C. After several washes in blocking solution, samples were incubated with appropriate secondary antibodies diluted in the blocking solution for 30 min at room temperature. Hoechst 33342 or DRAQ5 (ThermoFisher) were used to counterstain cell nuclei. Confocal images were acquired with Fluoview-10 software using an Olympus IX83 inverted microscope. Epifluorescence images were acquired with CellSense software using an Olympus IX83 microscope equipped with an Olympus DP80 dual CCD color and monochrome camera. Images were analyzed and prepared for publication using NIH ImageJ software.

## Myofiber size analysis

Myofiber size was measured by assessing 40X magnification confocal images of 2D and 3D cultures immunostained for sarcomeric α-actinin. 2D muscle culture images and flattened z-stack images of 3D muscle tissues were analyzed to quantify the diameter of each muscle fiber using the NIH ImageJ.

## Western blotting

3D tissues were collected at the indicated time points and flash frozen in liquid nitrogen, while 2D cell cultures were directly lysed in RIPA buffer (*Table 2*) and flash frozen. Processed 2D and 3D samples were stored at −80°C until all desired time points were collected. Tissues and 2D samples were lysed in RIPA buffer (ThermoFisher) containing protease inhibitors, and then lysates were analyzed

for total protein concentration using the BCA protein assay kit (ThermoFisher). 15 µg of protein was analyzed on an 8% SDS PAGE gel. Western blot was performed using a Bio-Rad Power Pac 1000 and Trans-Blot Turbo Transfer System to transfer the proteins from the polyacrylamide gel to a nitrocellulose membrane. Primary antibodies (*Table 3*) were incubated with membranes overnight at 4°C in milk-based blocking solution (*Table 2*). Membranes were washed 3 × 30 min with rocking in a Tris-buffered saline with Tween (TBST; *Table 2*) and then transferred into blocking solution containing horseradish peroxidase conjugated anti-rabbit and anti-mouse secondary antibodies (Cell Signaling; 1:5000). Chemoluminescence was performed using ECL substrate (ThermoFisher) with a MicroChemi 4.2 chemiluminescence imaging system (DNR Bio-Imaging Systems). Images were analyzed using the NIH ImageJ.

## AChR cluster analysis

α-bungarotoxin staining was performed to visualize and quantify the number, size, and morphology of AChR clusters in 2D cell and 3D tissue cultures. Briefly, fixed tissues were incubated with 5 nM of Alexa Fluor 647 conjugated α-bungarotoxin for 30 min to label AChRs. Samples were then washed with PBS and 40X images, all 0.1 $mm^2$ in area, were captured at a minimum of 6 random locations per sample. AChR cluster outlines in each 40X image were generated using the ImageJ particle analyzer. Clusters smaller than 5 $\mu m^2$ were excluded from analysis. AChR cluster outline drawings were binarized to facilitate downstream analysis. To assess cluster number, AChR cluster were quantified for each image and was then normalized to the number of sarcomeric α-actinin$^+$ fiber units present in the quantified image. To assess cluster area, the area of each individual AChR cluster was measured and averaged for each experiment using NIH ImageJ software. Fractal analysis was performed on α-bungarotoxin stained sample images to quantify AChR cluster morphological differences across the different culture conditions. After binarization of the AChR cluster outline drawings, lacunarity, a measure of gappiness and heterogeneity in a shape, was measured for each AChR cluster using the NIH ImageJ FracLac plug-in. The Sub Sample and Particle Analyzer method was used with FracLac along with default settings and four grid locations. Lacunarity was measured for each AChR cluster within an experiment and then averaged.

## Electrical stimulation

To ensure accurate and reproducible conditions for electrical stimulation, a custom-made stimulation chamber was produced using a 35 mm petri-dish, two carbon rods, and platinum wires. Before each use the stimulation chamber was sterilized using 70% ethanol. At day 14 of differentiation, an individual tissue was transferred to the chamber and covered in differentiation medium. Platinum wires were hooked up to a commercial function generator (Rigol DG1022U). A Rigol DS1102E digital oscilloscope was used to confirm the frequency and amplitude of signals before connecting the pulse generator to the platinum wires. 3D tissues were stimulated using square pulses with 20% duty cycle, 5V amplitude (field strength of 1.67 V/cm), and the reported frequencies.

## Calcium transient analysis

CD56$^+$ sorted human myogenic progenitor cells were transduced with a lentiviral vector encoding the fluorescent calcium indicator GCaMP6 driven by the muscle specific gene MHCK7 (AddGene plasmid #65042). Cells were then sorted to purify the infected cells based on GFP expression. Human skeletal muscle progenitor cultures expressing GCaMP6 were imaged using an Olympus IX83 microscope equipped with modules to control the temperature and $CO_2$ concentration. Videos were recorded at 4X magnification at 12 frames per second under physiological condition (37°C and 5% $CO_2$) in differentiation media using an Olympus DP80 dual CCD color and monochrome camera and CellSense software. Acetylcholine (BIO BASIC) was reconstituted to produce a 100 mM stock solution in PBS and was diluted to the final working concentration (as specified in the text) by addition directly into the culture chamber.

In glutamate stimulation studies, L-glutamate (Abcam) was first reconstituted to 100 mM in 1equal NaOH and then further diluted in HBSS (Gibco)/DMEM to produce a 100x stock solution (5 mM). For AChR epsilon subunit blocking studies, Waglerin-1 (Smartox Biotechnology) was prepared as a 100x stock solution in PBS and was added to cultures at a 1 µM working concentration 10 min prior to stimulation.

To assess the effect of Waglerin-1 on glutamate-stimulated calcium transients, a video was recorded for each tissue before and after glutamate stimulation. Video segments, equal in length, representing pre- and post-glutamate GcAMP6 signals were each projected into a 2D image. The 2D projected images were then subtracted to eliminate spontaneously active fibers from our analysis. Background from different imaging sessions were normalized. In *Figure 4C*, GCaMP6 signals were analyzed to quantify the area of glutamate responsive tissue at the same ROIs before (-) and after (+) Waglerin-1 treatment and presented as a fold-change. In *Figure 4D*, we identified all individual fibers that demonstrated GcAMP6 signal dampening in response to Waglerin treatment and then quantified signal in those fibers before (-) and then after (+) treatment and presented the data as a fold-change.

To assess the effect of BOTOX (Allergan, Irvine, CA) and d-tubocurarine (Sigma) treatments on glutamate-stimulated calcium transients, co-cultures were treated with BOTOX (1 U/ml) (*Santhanam et al., 2018*) and d-tubocurarine (25 µM) for at least 10 min before glutamate stimulation in their culture media. BOTOX was prepared at 100 U/ml in PBS and d-tubocurarine was reconstituted at 2.5 mM in DMEM.

## Tissue contraction quantification

To assess the contraction of neuromuscular tissues following glutamate and acetylcholine stimulations, co-cultures were stimulated under the indicated experimental conditions and videos were recorded at 4X magnification at 12 frames per second under physiological conditions (37°C and 5% $CO_2$). To quantify neuromuscular tissue contraction, videos were assembled into stacks using ImageJ software and 3 regions of interest were traced within each stack. Maximum movement distance for each trace was determined and averaged for each sample. Data are presented as movement (distance) in pixels.

## Length of functional connectivity between MN cluster and muscle fibers

In this studies, neuromuscular co-cultures were generated using GCaMP6 transduced human myogenic progenitor cells and a single motor neuron cluster. At week two of co-culture, tissues were stimulated by a 50 µM L-glutamate solution diluted in DMEM. Videos were captured at a frequency of 12 frames per second for at least 15 s before and after stimulation. Videos were processed exactly as described above (calcium transient analysis) to eliminate the spontaneously active fibers from the analysis. The location of the motor neuron cluster was identified from a bright field image, which was used to outline the structure with a circle in the epifluorescent images. Using NIH ImageJ software, concentric circles, each 100 µm larger in radius than the prior, were outlined around the motor neuron cluster until the circles encompassed all the glutamate responsive fibers (i.e. GcAMP6[+]) on the subtracted image. The number of active fibers in each circle were then quantified to determine the number of fibers in each concentric circle 'bin'. Binned data from three independent experiments were then reported on a histogram to report the average number of glutamate responsive fibers as it relates to the distance (i.e. concentric circle bin) from the motor neuron cluster.

## Electrophysiological recordings

Individual muscle fibers were impaled with 30–40 MΩ sharp electrodes pulled from borosilicate glass (World Precision Instruments), filled with 3M KCl. Membrane potential was recorded in the current clamp configuration using a Digidata 1440A and MultiClamp 700 A amplifier (Axon Instruments, Molecular Devices). Data were digitized at 10 kHz and filtered at 2.6 kHz. Data were quantified using MiniAnalysis (Synaptosoft). Each fiber was allowed to recover for a few minutes, to allow its resting membrane potential to stabilize before recordings were performed. For electrophysiological recordings following optogenetic stimulation, 3D muscle tissues were generated using human skeletal muscle progenitors transduced with a lentiviral vector encoding humanized ChR2 with H134R mutation fused to EYFP and driven by EF1a (AddGene plasmid #20942). Cells were sorted to purify the infected cells based on the EYFP signal. Optogenetic stimulation was performed using blue LED (KSL-70, RAPP OptoElectronic) with a wavelength of 470 nm, and controlled by the Axon amplifier software. In glutamate stimulation experiments, glutamate was pipetted by hand into the edge of

the bath and allowed to diffuse to the tissue. For all recordings, the bath solution was standard DMEM (Gibco).

**Remark:** To facilitate electrophysiological recordings, 3D neuromuscular tissues were removed from their culture well by popping the entire circular piece of PDMS containing the channel and co-culture tissue out of the 12-well plate and transferring it into a 35 mm culture dish. Addition of the neurotransmitter glutamate generates ripples and disturbs the equilibrium in the culture bath that may lead to the loss of sharp microelectrode recording. Therefore, the bolus of the neurotransmitter should be added from the side of the culture bath gently to then allow for its diffusion the tissue.

**Remark:** Spontaneous contractions of the neuromuscular tissues complicate the sharp microelectrode recordings. Neuromuscular tissues are most spontaneously active once removed from the incubator. As such, resting the tissue at room temperature for 5–10 min after transferring from the 37 C incubator reduces the spontaneous activity and facilitates the sharp microelectrode recordings.

## Myasthenia gravis disease modeling

Serum from three patients diagnosed with Anti-AChR MG (*Table 1*) was collected and IgG fractions were purified using a Protein A IgG purification kit (Thermofisher) based on the manufacturers instruction. Purified IgG was reconstituted in PBS and IgG content was quantified using a NanoDrop 1000 spectrophotometer (ThermoFisher). On Day 11 of neuromuscular co-culture, IgG was added to the differentiation media at 300 nM final concentration and 2% human serum (Sigma) was supplemented in the differentiation medium rather than horse serum. IgG from healthy human serum (Sigma) was used in 'healthy' control experiments. In these experiments, IgG was added once and the media was not exchanged thereafter. After 3 days of treatment, the Day 14 co-cultures were stimulated with glutamate followed by an acetylcholine stimulation and calcium transients were captured by imaging the GCaMP6 signals using an Olympus IX83 microscope. Video segments, equal in length, representing GCaMP6 signals from glutamate and ACh serially stimulated tissues were each projected into a 2D image. For *Figure 4—figure supplement 1B*, GCaMP6 signals were analyzed to quantify the area of ACh responsive tissue in equal sized ROI for healthy compared to MG IgG treated tissues, and presented as fold change. For *Figure 4G*, healthy and MG IgG treated tissues were analyzed to quantify the area of GCaMP6 signal at the same ROIs after glutamate (glut) and then acetylcholine (ACh) stimulation and the ratio of glutamate- to ACh-induced GCaMP6 signals was reported.

In a subset of experiments, we performed serial glutamate stimulation experiments in which we stimulated the same neuromuscular co-culture on Day 11 and then Day 14 of culture with a 50 µM L-glutamate solution and measured the area of GCaMP6$^+$ tissue as described above. Our analysis indicated that the area of glutamate responsive tissue increased from ~2% at Day 11 to more than 13% on Day 14, arguing against glutamate-induced cytotoxic effects arising from our experimental methods.

## FM 1–43 labeling and imaging

Motor neuron clusters were separated from undifferentiated single cells using Accutase (Thermo-Fisher) and transferred to a Geltrex coated 6-well plate on Day 14 of differentiation. Clusters were then cultured for an additional week in E6 media supplemented with 10 ng/mL BDNF and 10 ng/mL GDNF, to permit the regrowth of the neurites, and were then labeled with the FM 1–43 styryl dye (Molecular Probes) following the manufacturer's instructions. Briefly, MN clusters were stimulated using high potassium solution (60 mM) and incubated with FM 1–43 (2 µM) in HBSS (+Mg$^{2+}$- and + Ca$^{2+}$) for 20 min to enable dye loading. Clusters were washed with HBSS for at least one hour at room temperature before imaging. Samples were imaged using an IX83 Olympus confocal microscope with FV-10 software at physiological conditions (37°C and 5% CO$_2$). MN clusters were then stimulated with either high potassium solution (60 mM in PBS), L-glutamate (50 µM in HBSS), or control solutions (HBSS or PBS) while acquiring time-lapse video sequences. Videos were analysed for fluorescence intensity before and after stimulation at each indicated time-point using NIH ImageJ software.

## Gene expression analysis

Total RNA was extracted from three technical replicate muscle tissues or neuromuscular co-culture for each of 3 biological replicate experiments using the PureLink RNA Micro Kit according to the manufacturer's protocol (ThermoFisher). cDNA was reverse transcribed from 400 ng of RNA using the High-Capacity cDNA Reverse Transcription kit (Applied Biosystems). For quantitative real-time PCR (qRT-PCR), CHRNE and CHRNG primers were acquired from Bio-Rad and reactions were run according to manufacturer's protocol on the Roche LightCycler 480 (Roche) using LightCycler 480 SYBR Green I Master (Roche). All results were normalized to the housekeeping gene glyceraldehyde 3-phosphate dehydrogenase (GAPDH). Gene expression is reported in % of GAPDH expression ±SEM. To assess agrin gene expression in differentiated MNs, cDNA samples were prepared from three consecutive MN differentiations. Genes were amplified using Arktik thermal cycler according to the manufacturer's protocol (ThermoFisher). PCR amplification products were analyzed on a 2% agarose gel with SYBR safe DNA gel stain (Invitrogen). GAPDH gene expression served as the loading control. All oligo sequences are summarized in *Table 4*.

## Statistical analysis

Each study in this manuscript was performed using three primary myoblast lines derived from three separate muscle patient donors (N = 3 biological replicates). Each experiment within a study was set-up with cells from a separate muscle donor and included at least n = 3 technical replicates. Exceptions include *Figure 2I–J*, *Figure 4G*, *Figure 1—figure supplement 2G*, *Figure 1—figure supplement 3C*, and *Figure 4—figure supplement 1D*. In , two muscle tissue cultures (technical replicates) were treated with Agrin for each biological replicate (six samples in total). In *Figure 4G* and *Figure 4—figure supplement 1D*, IgG purified from sera collected from 3 MG patients was tested (N = 3 biological replicates), and compared to IgG purified from a single healthy donor and tested on 3D neuromuscular tissues engineered using a single immortalized cell line. In and 7 technical replicates from three muscle patient donors was analyzed at the 1 week culture time point. In *Figure 1—figure supplement 3C*, one muscle sample was analyzed for each muscle patient donor (N = 3 muscle patient donors). For all other neuromuscular co-culture studies, each primary myoblast line was co-cultured with MNs established from separate human pluripotent stem cell derivations.

Statistical analysis was performed on data obtained from technical replicates using GraphPad Prism 6.0 software. Statistical differences between experimental groups were determined in most studies by unpaired t-test. Exceptions to this are as follows: Two-way ANOVA followed by Tukey's and Sidak's multiple comparisons were performed in *Figure 1B* and *Figure 4—figure supplement 1B*. One-way ANOVA followed by Tukey's multiple comparisons was performed in *Figure 3B*, *Figure 3D*, *Figure 1—figure supplement 2G*, *Figure 1—figure supplement 3C*, and *Figure 3—figure supplement 2B*. Results are presented as mean ±SEM. p<0.05 was considered significant for all statistical tests. Absence of a significance symbol (*, #, $) indicates no significant differences.

## Acknowledgements

We thank Dr. Majid Ebrahimi and Dr. Louise Moyle for reviewing the manuscript and for providing critical feedback.

**Table 4.** Real-time PCR primer sequences.

| GENE | Species | Forward 5'−3' | Reverse 3'−5' |
|---|---|---|---|
| AGRN | human | CCTGACCCTCAGCTGGCCCT | AGATACCCAGGCAGGCGGCA |
| GAPDH | human | GTGAAGGTCGGAGTCAACG | TGAGGTCAATGAAGGGGTC |

DOI: https://doi.org/10.7554/eLife.44530.029

## Additional information

### Funding

| Funder | Grant reference number | Author |
|---|---|---|
| Natural Sciences and Engineering Research Council of Canada | CREATE TOeP | Mohsen Afshar Bakooshli |
| Krembil Foundation | | Mohsen Afshar Bakooshli |
| Toronto Musculoskeletal Centre | | Mohsen Afshar Bakooshli |
| University of Toronto | Ontario Graduate Scholarship | Mohsen Afshar Bakooshli |
| National Institutes of Health | 1F32NS083291-01A1 | Ethan S Lippmann |
| TD Bank Health Research Fellowship | | Ben Mulcahy |
| Natural Sciences and Engineering Research Council of Canada | | Bryan A Stewart |
| AFM-Téléthon | Myobank | Anne Bigot |
| AFM-Téléthon | Network of Genetic Biobanks (GTB12001D) | Elena Pegoraro |
| Natural Sciences and Engineering Research Council of Canada | RGPIN-2017-06738 | Mei Zhen |
| Canadian Institutes of Health Research | Foundation Scheme 15427 | Mei Zhen |
| Burroughs Wellcome Fund | Innovation in Regulatory Science Award | Randolph Scott Ashton |
| U.S. Environmental Protection Agency | STAR center grant 83573701 | Randolph Scott Ashton |
| National Institutes of Health | R21NS082618 | Randolph Scott Ashton |
| Toronto Western Arthritis Program | | Penney M Gilbert |
| Ontario Research Fund | 31390 | Penney M Gilbert |
| Canada Research Chairs | 950-231201 | Penney M Gilbert |
| Canada Foundation for Innovation | 31390 | Penney M Gilbert |
| Canada First Research Excellence Fund | 'Medicine by Design' OMNI-2017-01 | Penney M Gilbert |
| Ontario Institute for Regenerative Medicine | OMNI-2017-01 | Penney M Gilbert |
| University of Toronto Faculty of Medicine | Deans Fund | Penney M Gilbert |
| Natural Sciences and Engineering Research Council of Canada | RGPIN 435724-13 | Penney M Gilbert |

The funders had no role in study design, data collection and interpretation, or the decision to submit the work for publication.

### Author contributions

Mohsen Afshar Bakooshli, Christine T Nguyen, Conceptualization, Resources, Data curation, Formal analysis, Validation, Investigation, Visualization, Methodology, Writing—original draft, Writing—review and editing; Ethan S Lippmann, Resources, Formal analysis, Validation, Investigation,

Methodology, Writing—original draft, Writing—review and editing; Ben Mulcahy, Resources, Data curation, Formal analysis, Validation, Investigation, Visualization, Methodology, Writing—original draft, Writing—review and editing; Nisha Iyer, Hubrecht van den Dorpel, Resources, Formal analysis, Validation, Methodology, Writing—review and editing; Kayee Tung, Project administration; Bryan A Stewart, Conceptualization, Resources, Formal analysis, Supervision, Funding acquisition, Validation, Investigation, Methodology, Writing—original draft, Writing—review and editing; Tobias Fuehrmann, Conceptualization, Resources, Formal analysis, Supervision, Validation, Investigation, Methodology, Writing—review and editing; Molly Shoichet, Resources, Formal analysis, Supervision, Funding acquisition, Writing—review and editing; Anne Bigot, Henry Ahn, Howard Ginsberg, Resources, Formal analysis, Writing—review and editing; Elena Pegoraro, Resources, Formal analysis, Methodology, Writing—review and editing; Mei Zhen, Conceptualization, Resources, Formal analysis, Supervision, Funding acquisition, Validation, Methodology, Writing—review and editing; Randolph Scott Ashton, Conceptualization, Resources, Data curation, Formal analysis, Supervision, Funding acquisition, Validation, Methodology, Writing—original draft, Project administration, Writing—review and editing; Penney M Gilbert, Conceptualization, Resources, Data curation, Formal analysis, Supervision, Funding acquisition, Validation, Investigation, Visualization, Methodology, Writing—original draft, Project administration, Writing—review and editing

## Author ORCIDs
Ben Mulcahy  http://orcid.org/0000-0002-3336-245X
Bryan A Stewart  http://orcid.org/0000-0003-2520-3632
Mei Zhen  http://orcid.org/0000-0003-0086-9622
Randolph Scott Ashton  http://orcid.org/0000-0002-6842-7022
Penney M Gilbert  http://orcid.org/0000-0001-5509-9616

## Ethics
Human subjects: Our study made use of skeletal muscle tissue that was removed in the course of scheduled surgical procedures at St. Michael's Hospital and would have otherwise been discarded as surgical waste. Informed consent and consent to publish was obtained from all patients whose tissue was used in this study. This study was reviewed and approved by the St. Michael's Research Ethics Board (REB# 13-370). The University of Toronto Office of Research Ethics performed a second review of the approved REB and granted administrative approval (Protocol#30754). Our study used sera collected from patients with myasthenia gravis with consent and approval to publish via the Telethon Network of Genetic Biobanks.

## Decision letter and Author response
Decision letter https://doi.org/10.7554/eLife.44530.032
Author response https://doi.org/10.7554/eLife.44530.033

## Additional files
### Supplementary files
• Transparent reporting form
DOI: https://doi.org/10.7554/eLife.44530.030

### Data availability
All data supporting the findings of this study are available within the article.

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
