## [Decision Letter]

Thank you for submitting your article "A 3D culture model of innervated human skeletal muscle enables studies of the adult neuromuscular junction" for consideration by *eLife*. Your article has been reviewed by three peer reviewers, including Andrew Brack as the Reviewing Editor and Reviewer #1, and the evaluation has been overseen by Marianne Bronner as the Senior Editor.

The reviewers have discussed the reviews with one another and the Reviewing Editor has drafted this decision to help you prepare a revised submission.

Summary:

The manuscript from the Gilbert lab provides the first demonstration of a robust method to generate mature muscle fibers with functioning 'adult' NMJ in vitro. This has been a significant hurdle for many years. This new approach will have significant utility for basic scientists working in the muscle or NMJ field, and those that model disease in vitro. The experiments have been performed to the highest standard, the data are compelling, the methods thorough, and the statistics appropriate. This work is a significant technical advance.

Reviewers felt that the authors provide compelling evidence for the use of 3D muscle cultures. While there is general excitement for the manuscript, there were two main issues raised. 1) How easy is it to adopt this technique? It was mentioned that readers would benefit from a more transparent discussion on the method, outlining the difficult steps and the pitfalls that drive its failure/success. 2) Further evidence to describe the maturation status of the NMJs. Is activity required for the maturation of the muscle fibers?

Below are the main points to consider. These points are given more context within the reviewer's individual comments.

1) How easy is this method to be adopted? Please provide information outlining the ease of adaptability and the technical hurdles that are critical for success.

2) More data to determine the maturation of the NMJs:

- The authors should stain individual NMJs in their 3D system with antibodies against the other proteins (e.g., rapsyn, MuSK) that they study in bulk cultures. Do they get more fully formed NMJs?

- Is there any difference in the NMJs between two systems when investigated more at the level of single NMJs (which could be done using antibody staining along with α-bungarotoxin)?

- Does curare block all of the activity?

- The increase in EPP frequency is not obvious. The authors should apply TTX to the cultures to at least distinguish between EPPs and mEPPS. Do the authors see action potentials prior to glutamate application?

3) Are there lessons that can be learnt from the 3D cultures that could be implemented to improve 2D cultures? If the authors have data that could be incorporated within the 2-month revision window, please do so. If not, the manuscript would benefit from a discussion on this issue.

Reviewer #1:

The manuscript from the Gilbert lab provides the first demonstration of a robust method to generate mature muscle fibers with functioning 'adult' NMJ in vitro. This has been a significant hurdle for many years. This new approach will have significant utility for basic scientists working in the muscle or NMJ field, and those that model disease in vitro. The experiments have been performed to the highest standard, the data are compelling, the methods thorough, and the statistics appropriate. This work is a significant technical advance.

I have one request for the authors, which stems from the fact that the take away message is the ease of the method. And while the data support this notion, I would ask the reviewers to provide more information (in commentary or data) as to where the limitations or difficulties in the method lie. For example, what is the day-to-day variability between neuromuscular samples, or the variability between dishes? Is there uniformity across time/samples/experimentalist?

In essence, it would be helpful to any adopter of this method if they understood the limitations and potential roadblocks.

Reviewer #2:

The paper by Bakooshli et al. presents a new system to study human muscle development, neuromuscular junction (NMJ) formation, and, perhaps, neuromuscular disorders. The paper is fairly comprehensive and mostly quite well-done. I think it is suitable for *eLife* and should be published after suitable revisions.

I have three general comments and then several specific questions detailed below.

First, the authors have compelling data illustrating differences in the speed of muscle development. Can any lessons be learned (see also below) from 3D that could be applied to 2D cultures?

Second, while the authors do an admirable job of comparing their new 3D systems with a more conventional 2D systems (always difficult to do fairly since there is always a clear preference for one vs. the author), they should try to objectively present the advantages AND disadvantages of 3D. First of all, it's not clear to me how easy the system described in this manuscript would be to establish in a non-engineering based lab. A guide to its use or an outline of the type of future developments that would make this system more routine would be useful. Many of the authors' measurements are made on "bulk" cultures, with fewer being done on individual fibers or individual NMJ's. So, certainly one downside may be that 2D does make some of that more straightforward (many of my questions listed below relate to what can be observed at a single fiber level in 3D and a few require additional straightforward experiments.)

My final general comment is that the myasthenia gravis work seems like an almost unnecessary add-on. Was this done to compare the current system to that used by Steinbeck et al., 2016?

Reviewer #3:

In vitro studies of skeletal muscle can be problematic because the in vivo function of these cells requires interaction with multiple other cell types including motor neurons, however these interactions are difficult to replicate ex vivo. This article describes a method for generating de novo differentiated muscle fibers 3D culture in vitro, coupled with generation of pluripotent stem cell-derived motor neurons, which will self-assemble into functional units in coculture. To validate the technique, the authors showed that these muscle fibers express myosin heavy chain and acetylcholine receptor isoforms specific to mature adult muscle (which are not usually upregulated in vitro) and demonstrated that the neuromuscular synapses (NMJs) respond in a physiologically appropriate fashion to electrical and chemical stimulation, as measured by muscle contractile activity, calcium flux, and electrophysiology. These muscle fibers and motor neurons were generated from human cells, so as a final proof-of-principle the authors treated the matured neuromuscular units with serum from patients with myasthenia gravis (MG), an autoimmune disease which targets the mature isoform of NMJ receptors and diminishes their function. Within two weeks of treatment, they observed alterations in the in vitro generated NMJs mimicking clinical signs in patients, supporting the potential utility of these in vitro cultures for modeling human disease.

This paper is a fairly straightforward paper; the experiments are well designed and documented and the paper itself is clearly written. The authors are correct in stating that there is an unmet need in the field for an in vitro model of the mature NMJ, and the culture system they describe is better than anything currently in use in the field. As this is primarily a technique paper, I don't think additional studies beyond what is already included are necessary, although there are several that could be done which come to mind- if this is published, I think it will be of significant use to the field and those experiments will be carried out by this group and others.

---

## [Author Response]

Reviewers felt that the authors provide compelling evidence for the use of 3D muscle cultures. While there is general excitement for the manuscript, there were two main issues raised. 1) How easy is it to adopt this technique? It was mentioned that readers would benefit from a more transparent discussion on the method, outlining the difficult steps and the pitfalls that drive its failure/success. 2) Further evidence to describe the maturation status of the NMJs. Is activity required for the maturation of the muscle fibers?Below are the main points to consider. These points are given more context within the reviewer's individual comments.1) How easy is this method to be adopted? Please provide information outlining the ease of adaptability and the technical hurdles that are critical for success.

We thank the reviewers for the opportunity to provide more detail related to our personal experiences with this co-culture method and platform. Indeed, this information is key to ensuring widespread adoption, together with our proactive efforts in hosting labs from across the world for hands-on training. We now provide additional detail in the Materials and methods and we expand on the point of technical hurdles and system adaptation pointers in the Discussion.

See also detailed response to reviewer #1.

2) More data to determine the maturation of the NMJs:- The authors should stain individual NMJs in their 3D system with antibodies against the other proteins (e.g., rapsyn, MuSK) that they study in bulk cultures. Do they get more fully formed NMJs?- Is there any difference in the NMJs between two systems when investigated more at the level of single NMJs (which could be done using antibody staining along with α-bungarotoxin)?

To address this issue, we set up cultures and performed immunostaining on 2D and 3D muscle alone cultures as well as 2D and 3D neuromuscular co-cultures. We did not observe any examples of AChR clusters co-localizing with either rapsyn or MuSK proteins in our 2D muscle-alone or neuromuscular cultures (Figure 2—figure supplement 2C).

We observed a single incidence of MuSK protein co-localization with an AChR cluster in our 3D muscle-alone cultures (Figure 2—figure supplement 2D) and no incidences of rapsyn co-localization with AChR clusters in our 3D muscle-alone cultures.

By comparison, the prevalence of rapsyn and MuSK co-localizing with bungarotoxin stained AChR clusters was substantially higher in 3D neuromuscular co-cultures. Two representative examples where we observed co-localization of rapsyn (Figure 2D) and MuSK proteins (Figure 2—figure supplement 2E) with MN neurites at bungarotoxin stained AChR cluster sites are shown. However, we note that not all the AChR clusters co-localized with MuSK and rapsyn in 3D neuromuscular co-cultures at the experimental time point we evaluated (Day 12).

- Does curare block all of the activity?

We use 25 μM treatment of d-tubocurarine in our system. At this concentration we observe over 99% blocking of changes in muscle fiber calcium transients post glutamate stimulation of motor neurons. This is matched with the results from Ko et al., 2019 were they observe a full block at 50 μM. In addition, we do not observe a change in calcium transients of d-tubocurarine treated (25 μM) myofibers stimulated with 100 μM, ACh (data not shown) which demonstrates the potency of d-tubocurarine blocking at this concentration. These data are consistent with previous reports (Madden et al., 2015) using similar concentration of curare and recent work from our laboratory deposited on BioRxiv (https://doi.org/10.1101/562819). We now make this point and refer to these citations in the text.

- The increase in EPP frequency is not obvious. The authors should apply TTX to the cultures to at least distinguish between EPPs and mEPPS. Do the authors see action potentials prior to glutamate application?

As (correctly) alluded to by all reviewers, we faced challenges in performing electrophysiological recordings. We did indeed set up neuromuscular co-cultures to address this concern, but in the end, ran into technical hurdles that precluded the gathering of data. As the protocol requires 5 weeks from beginning to end, we faced some pressure in collecting the data in the 2-month window, but we did try. As suggested by all reviewers, we now make clear in the Discussion challenges associated with performing e-phys in culture and offer suggestions to adapt methods for future users who wish to perform e-phys in this system.

We have updated the text to faithfully reflect the data we have in hand. Specifically, we never saw spontaneous activity in 3D muscle-alone recordings (Figure 1—figure supplement 3E), whereas, it was commonplace to capture spontaneous activity in 3D neuromuscular co-cultures (Figure 3—figure supplement 2C). We now speculate (rather than conclude), that these spontaneous endogenous end plate potentials (EPPs) are caused by spontaneous release of ACh from the motor neuron axon terminals at the NMJs, and in the Discussion we acknowledge that without performing blocking studies, we cannot make this conclusion definitively. While our analysis indicated a glutamate evoked change in EPP frequency, we understand that the trace provided was less than satisfactory. We now provide more representative traces we collected in the course of studies aimed at capturing evoked potentials using e-phys (Figure 3—figure supplement 2C). In the Discussion we acknowledge that without performing blocking studies, we cannot make this conclusion definitively.

Action potential-like events were only observed post glutamate stimulation. We now make this point in the Results section.

3) Are there lessons that can be learnt from the 3D cultures that could be implemented to improve 2D cultures? If the authors have data that could be incorporated within the 2-month revision window, please do so. If not, the manuscript would benefit from a discussion on this issue.

We now address this point in our Discussion. See corresponding text below.

Based on our comparative analysis, a major advantage of a 3D culture system is the ability to support the contractility of maturing myofibers, which then are the ideal template for motor neuron innervation in as early as 12 days of differentiation. In addition, myofibers form packed and aligned structures in a 3D system which is not observed in standard tissue culture plastic 2D cultures.

As such, we expect that focusing efforts on modifying 2D cultures to control the microenvironment in ways that can accommodate myofiber contractility and myofiber alignment, might result in the formation of functional NMJs in vitro at earlier time-points, and perhaps even support the de novo embryonic to adult switch in AChR subunit type. Indeed, a recent report by Ko and colleagues (Ko et al., 2019) working with rodent myoblast and neural cells highlights a role for matrix topography and myofiber alignment as it relates to innervation efficiency in 2D culture, though more comparative analysis is required to understand how this type of approach impacts human NMJ formation in culture as compared to a 3D co-culture system.

Reviewer #1:[…] I have one request for the authors, which stems from the fact that the take away message is the ease of the method. And while the data support this notion, I would ask the reviewers to provide more information (in commentary or data) as to where the limitations or difficulties in the method lie. For example, what is the day-to-day variability between neuromuscular samples, or the variability between dishes? Is there uniformity across time/samples/experimentalist?In essence, it would be helpful to any adopter of this method if they understood the limitations and potential roadblocks.

We thank the reviewer for highlighting this point. Here we provide additional detail to address the issue of technical challenges and reproducibility. We have added text to the Materials and methods to provide tips for success. We also expand on technical limitations and suggest adaptations that will expand the usability of the system.

1) While the muscle culture protocol consistently yields muscle tissues, the use of Velcro as an anchor points is not ideal. Glue-ing the Velcro into each side of the culture depression is time consuming and requires some practice. In some instances, for example if the Velcro hooks were broken and/or weak, a 3D muscle tissue might rupture at an early time point. As a result, the user must assess the quality of the anchor points thoroughly to ensure the successful formation and maintenance of the muscle tissues. To overcome this technical issue, we designed and produced a 96 well plate that enables reproducible bulk production of muscle microtissues in vitro that uses deflective rubber posts cast directly into the plate as ‘tendon’ anchor points, rather than Velcro (BioRxiv,: https://doi.org/10.1101/562819).

2) Challenges associated with specifying pluripotent stem cells to the motor neuron lineage can delay planned experiments. This is a particular hurdle given that current motor neuron specification protocols are 2-3 weeks long. Common issues include the batch to batch variability and/or limited shelf-life of the motor neuron differentiation factors (retinoic acid, purmorphamine, sonic hedgehog, and DAPT) which can have a significant impact on the success of the motor neuron differentiation protocol. The initial seeding density of Olig2^+^ progenitor cells also impacts motor neuron cluster formation and differentiation yields. As such, potential adaptors need to test each new batch of differentiation factors and also ensure consistent seeding criteria of progenitor cells to ensure high differentiation yields. Finally, it should be noted that excessive shearing of Olig2^+^ cells upon thawing and plating, reduces the differentiation efficiency in subsequent steps.

3) The co-culture protocol relies on collecting individual motor neuron clusters to mix together with muscle progenitor cells, as explained in the Materials and the methods. This is advantageous as it enables the user to collect a defined number of uniformly sized clusters that are free of any contaminating, undifferentiated cells. However, this step is very labor intensive and limits the throughput of the co-culture tissue production. Therefore, future work aimed at defining protocols for enzymatic collection of clusters followed by a stepwise straining process to support high throughput collection of uniformly sized motor neuron clusters is needed.

4) To ensure easy fabrication of neuromuscular co-culture tissues, motor neuron clusters are mixed directly into the hydrogel mix containing the mononucleated muscle progenitor cells. While this allows for easy fabrication of neuromuscular tissues, the process precludes control over the final localization of motor neuron clusters as the neuromuscular tissues form. As such, the proximity of motor neuron clusters to one another, and the degree of innervation within co-culture tissues can vary a bit from one tissue to another.

5) Compared to 3D muscle only tissues, we observed a lot of contractile activity in neuromuscular co-cultures, which we expect is due to the spontaneous firing of motor neuron cells present in co-cultures. Based on the literature, we surmise that the higher motor neuron induced muscle fiber activity contributes to the advanced contractile maturation of muscle fibers in our co-cultures (aligned with previous reports Akaaboune et al., 1999; Andreose et al., 1993; Caroni et al., 1993; Skorpen et al., 1999), but in turn also led to higher rates of muscle rupture in co-cultures. Based on a recent report (Cvetkovic et al., 2017) highlighting the influence of plasmin, cathepsin L, MMP-2, and MMP-9 proteases on in vitro muscle tissue longevity, we propose that using an array of protease inhibitors, in addition to ACA, might aid in preventing contraction-induced neuromuscular co-culture demise.

6) In general, analyses at the single fiber level require the user to develop tools or adapt the protocol to improve feasibility. For example, electrophysiological recordings in the co-culture system are feasible, but highly challenging for a number of reasons. First, spontaneous or induced muscle tissue contractions in 3D neuromuscular co-cultures frequently resulted in the loss of pipette contact with the cell membrane during recordings. As a result, recording where glutamate is added to the culture bath as a stimulation method, which elicits tissue movement via multiple muscle fibers contracting in unison (as seen in Video 7) were exceedingly challenging and as such, only successful in few events, as reported in the manuscript. However, this challenge can be overcome by performing targeted stimulation of single motor neurons with electrical or neurotransmitter stimulation, or by stimulating with blue light in the case of motor neurons genetically modified to express a light sensitive channel (e.g.) channelrhodopsin. In addition, identifying innervated muscle fibers is challenging. Using fluorescently labelled muscle cells (e.g. GCaPM6+, membrane anchored fluorophore) and motor neuron (e.g. HB9-GFP, mCherry, neurofilament-GFP) cells will dramatically improve the success rate in identifying innervated muscle fibers in 3D neuromuscular co-cultures. Notably, the 3D nature of the co-culture system reduces the incidence of pipette breakage common in studies of plastic cultured myotubes. However, without added myotube maturation through contraction regimes or otherwise, it should be noted that myotubes in culture are somewhat smaller than those in adult animals, which can introduce some difficulty in recording. Other types of single fiber or single molecule analyses, such as immunostaining or time-lapse microscopy to follow a single acetylcholine receptor subunit, are more challenging in 3D as compared to 2D culture, but are feasible.

Reviewer #2:The paper by Bakooshli et al. presents a new system to study human muscle development, neuromuscular junction (NMJ) formation, and, perhaps, neuromuscular disorders. The paper is fairly comprehensive and mostly quite well-done. I think it is suitable for eLife and should be published after suitable revisions.

We thank the reviewer for acknowledging the value and quality of our work.

I have three general comments and then several specific questions detailed below.First, the authors have compelling data illustrating differences in the speed of muscle development. Can any lessons be learned (see also below) from 3D that could be applied to 2D cultures?

We now address this point in our Discussion. See corresponding text below.

Based on our comparative analysis, a major advantage of a 3D culture system is the ability to support the contractility of maturing myofibers, which then are the ideal template for motor neuron innervation in as early as 12 days of differentiation. In addition, myofibers form packed and aligned structures in a 3D system which is not observed in standard tissue culture plastic 2D cultures.

As such, we expect that focusing efforts on modifying 2D cultures to control the microenvironment in ways that can accommodate myofiber contractility and myofiber alignment, might result in the formation of functional NMJs in vitro at earlier time-points, and perhaps even support the de novo embryonic to adult switch in AChR subunit type. Indeed, a recent report by Ko and colleagues (Ko et al., 2019) working with rodent myoblast and neural cells highlights a role for matrix topography and myofiber alignment as it relates to innervation efficiency in 2D culture, though more comparative analysis is required to understand how this type of approach impacts human NMJ formation in culture as compared to a 3D co-culture system.

Second, while the authors do an admirable job of comparing their new 3D systems with a more conventional 2D systems (always difficult to do fairly since there is always a clear preference for one vs. the author), they should try to objectively present the advantages AND disadvantages of 3D. First of all, it's not clear to me how easy the system described in this manuscript would be to establish in a non-engineering based lab. A guide to its use or an outline of the type of future developments that would make this system more routine would be useful.

The 3D system culture platform requires a laser engraver (cutter) to prepare the dog-bone shaped molds to fabricate rubber channels in a standard 12-well plate. Otherwise, no specialized equipment or know-how is required. A detailed experimental procedure about the fabrication of the platform is provided in the Materials and methods section under the “PDMS mold fabrication for 3D human muscle tissue culture”.

Many of the authors' measurements are made on "bulk" cultures, with fewer being done on individual fibers or individual NMJ's. So, certainly one downside may be that 2D does make some of that more straightforward (many of my questions listed below relate to what can be observed at a single fiber level in 3D and a few require additional straightforward experiments.)

While we highlight the advantages of the 3D culture compared to the conventional 2D system in our manuscript, the reviewer is undoubtedly right that certain aspects of muscle studies are more straightforward when done in a 2D culture. For example, image based analyses are often easier to perform in 2D systems.

Another challenge with our 3D co-culture studies, as we elaborate on above, was to locate the individual fibers for electrophysiological recordings. Using fluorescently labelled cells would help to address this issue and would facilitate individual fiber analysis in a 3D system. Further, stimulating motor neurons locally using electrical, chemical, or optogenetic methods, as opposed to adding glutamate directly to the bath (as done in our studies), will mitigate issues of tissue movement that often accompanied lost recording opportunities.

Despite these challenges, it is important to note that the development of NMJ formation is a time taking process which, by definition, includes formation of contractile myofibers and AChR clustering. Therefore, a 2D or 3D system capable of myofiber maintenance and contractile maturation is needed to study the formation of functional adult NMJs.

My final general comment is that the myasthenia gravis work seems like an almost unnecessary add-on. Was this done to compare the current system to that used by Steinbeck et al., 2016?

We acknowledge and understand this comment about our myasthenia gravis work. As one might expect, these experiments were included to appease a prior reviewer. We agree that it does not substantially add to the primary message of the manuscript, but as it does demonstrate the ease of modeling an autoimmune NMJ disease in the in vitro culture system in as early as 14 days of differentiation, we prefer that the work remain in the manuscript.